# A Linearly Convergent Method for Non-Smooth Non-Convex Optimization on the Grassmannian with Applications to Robust Subspace and Dictionary Learning

**Zhihui Zhu**
MINDS
Johns Hopkins University
`zzhu29@jhu.edu`

**Tianyu Ding**
AMS
Johns Hopkins University
`tding1@jhu.edu`

**Manolis C. Tsakiris**
SIST
ShanghaiTech University
`mtsakiris@shanghaitech.edu.cn`

**Daniel P. Robinson**
ISE
Lehigh University
`daniel.p.robinson@lehigh.edu`

**René Vidal**
MINDS
Johns Hopkins University
`rvidal@jhu.edu`

## Abstract

Minimizing a non-smooth function over the Grassmannian appears in many applications in machine learning. In this paper we show that if the objective satisfies a certain Riemannian regularity condition (RRC) with respect to some point in the Grassmannian, then a projected Riemannian subgradient method with appropriate initialization and geometrically diminishing step size converges at a linear rate to that point. We show that for both the robust subspace learning method Dual Principal Component Pursuit (DPCP) and the Orthogonal Dictionary Learning (ODL) problem, the RRC is satisfied with respect to appropriate points of interest, namely the subspace orthogonal to the sought subspace for DPCP and the orthonormal dictionary atoms for ODL. Consequently, we obtain in a unified framework significant improvements for the convergence theory of both methods.

## 1   Introduction

Optimization problems on the Grassmannian $\mathbb{G}(c, D)$ (a.k.a. the Grassmann manifold that consists of the set of linear $c$-dimensional subspaces in $\mathbb{R}^D$), such as principal component analysis (PCA), appear in a wide variety of applications including subspace tracking [3], system identification [43], action recognition [36], object categorization [20], dictionary learning [34, 39], robust subspace recovery [26, 42], subspace clustering [41], and blind deconvolution [50]. However, a key challenge is that the associated optimization problems are often non-convex since the Grassmannian is a non-convex set.

One approach to solving optimization problems on the Grassmanian is to exploit the fact that the Grassmannian is a Riemannian manifold and develop generic Riemannian optimization techniques. When the objective function is twice differentiable, [4] shows that Riemannian gradient descent and Riemannian trust-region methods converge to first- and second-order stationary solutions, respectively. When Riemannian gradient descent is randomly initialized, [23] further shows that it converges to a second-order stationary solution almost surely, but without any guarantee on the convergence rate. Non-smooth trust region algorithms [19], gradient sampling methods [9, 8], and proximal gradient methods [7] have also been proposed for non-smooth manifold optimization when the objective function is not continuously differentiable. However, the available theoretical results establish

convergence to stationary points from an arbitrary initialization with either no rate of convergence guarantee, or at best a sublinear rate.[1]

On the other hand, when the constraint set is convex, [11, 10, 27] show that subgradient methods can handle non-smooth and non-convex objective functions as long as the problem satisfies certain regularity conditions called *sharpness* and *weak convexity*. In such a case, R-linear convergence[1] is guaranteed (e.g., see robust phase retrieval [13] and robust low-rank matrix recovery [27]). Analogous to other regularity conditions for smooth problems, such as the regularity condition of [6] and the error bound condition in [29], sharpness and weak convexity capture regularity properties of non-convex and non-smooth optimization problems. However, these two properties have not yet been exploited for solving problems on the Grassmannian, or other non-convex manifolds.

A related regularity condition, which in this paper is called the Riemannian Regularity Condition (RRC), has already been exploited for orthogonal dictionary learning (ODL) [2], which solves an $\ell_1$ minimization problem on the sphere, a manifold parameterizing $\mathbb{G}(1, D)$. However, under this RRC, projected Riemannian subgradient methods have only been proved to converge at a *sublinear* rate. On the other hand, a projected subgradient method has been successfully used and proved to converge at a *piecewise linear* rate for Dual Principal Component Pursuit (DPCP) [42, 51], a method that fits a linear subspace to data corrupted by outliers. However, i) the convergence analysis does not reveal where the improvement in the convergence rate comes from and ii) is restricted to optimization on the sphere ($\mathbb{G}(1, D)$) even for subspaces of codimension higher than one.

In this paper we make the following specific contributions:

- In Theorem 1 we prove that the projected Riemannian subgradient method for the Grassmannian (Algorithm 1), with an appropriate initialization and geometrically diminishing step size, converges to a point of interest at an *R-linear rate* if the problem satisfies the RRC (Definition 1). The RRC characterizes a local geometric property of the Grassmannian-constrained non-convex and non-smooth problem relative to a point of interest. Informally, the RRC requires that, in the neighborhood of the point of interest, the negative of the Riemannian subgradient should have a sufficiently small angle with the direction pointing toward the point of interest (Figure 1).

- We prove that the optimization problem associated with DPCP satisfies the RRC, which allows us to apply our new result and conclude that the projected Riemannian subgradient method converges at an R-linear rate to a basis for the orthogonal complement of the underlying $(D - c)$-dimensional linear subspace. This is the first result to extend previous guarantees [42, 51, 12] from codimension 1 to higher codimensions, enabling us to efficiently find the entire orthogonal basis by solving the learning problem directly on $\mathbb{G}(c, D)$, as opposed to the less efficient approach of solving a sequence of $c$ problems on $\mathbb{G}(1, D)$[42]. Even for subspaces of codimension 1 (i.e., hyperplanes), our result improves upon [51] by allowing for (i) a much simpler step size selection strategy that requires little fine-tuning, and (ii) a weaker condition on the required initialization.

- Together with the already established RRC for ODL in [2], our new result implies that the projected Riemannian subgradient method converges at an R-linear rate to atoms of the underlying dictionary, thus improving upon [2], which established only a sublinear convergence rate.

## 2 Background and Notation

In this paper, we consider minimization problems on the Grassmannian $\mathbb{G}(c, D)$. For computations, it is desirable to parameterize points on the Grassmannian. An element of $\mathbb{G}(c, D)$ can be represented by an orthonormal matrix in $\mathbb{O}(c, D) =: \{\boldsymbol{B} \in \mathbb{R}^{D \times c} : \boldsymbol{B}^\top \boldsymbol{B} = \mathbf{I}_c\}$, which is the well-known Stiefel manifold. When $D = c$, we denote $\mathbb{O}(c, c)$ by $\mathbb{O}(c)$, the orthogonal group. This matrix representation is not unique since $\text{Span}(\boldsymbol{B}\boldsymbol{Q}) = \text{Span}(\boldsymbol{B})$ for any $\boldsymbol{Q} \in \mathbb{O}(c)$. Thus, we say $\boldsymbol{A} \in \mathbb{G}(c, D)$ is equivalent to $\boldsymbol{B}$ if $\text{Span}(\boldsymbol{A}) = \text{Span}(\boldsymbol{B})$. With this understanding, we use $\boldsymbol{B}$ to represent the equivalence class $[\boldsymbol{B}] = \{\boldsymbol{B}\boldsymbol{Q} : \boldsymbol{Q} \in \mathbb{O}(c)\}$ and consider the parameterized problem [14, 20]

$$\underset{\boldsymbol{B} \in \mathbb{O}(c,D)}{\text{minimize}} f(\boldsymbol{B}), \tag{1}$$

where $f : \mathbb{R}^{D \times c} \to \mathbb{R}$ is locally Lipschitz, possibly non-convex and non-smooth, and invariant to the action of $\mathbb{O}(c)$, i.e., $f(\boldsymbol{B}) = f(\boldsymbol{BQ})$ for any $\boldsymbol{Q} \in \mathbb{O}(c)$. Again, the global minimum of (1) is not unique as if $\boldsymbol{B}^{\star}$ is a global minimum, then any point in $[\boldsymbol{B}^{\star}]$ is also a global minimum.

For any $\boldsymbol{A}, \boldsymbol{B} \in \mathbb{O}(c, D)$, the principal angles between $\mathrm{Span}(\boldsymbol{A})$ and $\mathrm{Span}(\boldsymbol{B})$ are defined as [38] $\phi_i(\boldsymbol{A}, \boldsymbol{B}) = \arccos(\sigma_i(\boldsymbol{A}^{\top}\boldsymbol{B}))$ for $i = 1, \ldots, c$, where $\sigma_i(\cdot)$ denotes the $i$-th singular value. We then define the distance between $\boldsymbol{A}$ and $\boldsymbol{B}$ as

$$\mathrm{dist}(\boldsymbol{A}, \boldsymbol{B}) := \sqrt{2 \sum_{i=1}^{c} \big(1 - \cos(\phi_i(\boldsymbol{A}, \boldsymbol{B}))\big)} = \min_{\boldsymbol{Q} \in \mathbb{O}(c)} \|\boldsymbol{B} - \boldsymbol{AQ}\|_F, \quad (2)$$

where the last term is also known as the *orthogonal Procrustes problem*. The last equality in (2) follows from the result [21] according to which the optimal rotation matrix $\boldsymbol{Q}$ minimizing $\|\boldsymbol{B} - \boldsymbol{AQ}\|_F$ is $\boldsymbol{Q}^{\star} = \boldsymbol{UV}^{\top}$, where $\boldsymbol{U\Sigma V}^{\top}$ is the SVD of $\boldsymbol{A}^{\top}\boldsymbol{B}$. Thus, $\mathrm{dist}(\boldsymbol{A}, \boldsymbol{B}) = 0$ iff $\mathrm{Span}(\boldsymbol{A}) = \mathrm{Span}(\boldsymbol{B})$. We also define the projection of $\boldsymbol{B}$ onto $[\boldsymbol{A}]$ as

$$\mathcal{P}_{\boldsymbol{A}}(\boldsymbol{B}) = \boldsymbol{AQ}^{\star}, \quad \text{where} \quad \boldsymbol{Q}^{\star} = \arg \min_{\boldsymbol{Q} \in \mathbb{O}(c)} \|\boldsymbol{B} - \boldsymbol{AQ}\|_F. \quad (3)$$

Here, $\boldsymbol{AQ}^{\star}$ is in $[\boldsymbol{A}]$, with $\boldsymbol{Q}^{\star}$ representing a nonlinear transformation of $\boldsymbol{A}^{\top}\boldsymbol{B}$, as described above.

Since $f$ can be non-smooth and non-convex, we utilize the Clarke subdifferential, which generalizes the gradient for smooth functions and the subdifferential in convex analysis. The Clarke subdifferential of a locally Lipschitz function $f$ at $\boldsymbol{B}$ is defined as [2]

$$\partial f(\boldsymbol{B}) := \mathrm{conv} \left\{ \lim_{i \to \infty} \nabla f(\boldsymbol{B}_i) : \boldsymbol{B}_i \to \boldsymbol{B}, f \text{ differentiable at } \boldsymbol{B}_i \right\},$$

where conv denotes the convex hull. When $f$ is differentiable at $\boldsymbol{B}$, its Clarke subdifferential is simply $\{\nabla f(\boldsymbol{B})\}$. When $f$ is not differentiable at $\boldsymbol{B}$, the Clarke subdifferential is the convex hull of the limit of gradients taken at differentiable points. Note that the Clarke subdifferential $\partial f(\boldsymbol{B})$ is a nonempty and convex set since a locally Lipschitz function is differentiable almost everywhere.

Since we consider problems on the Grassmannian, we use tools from Riemannian geometry to state optimality conditions. From [14], the tangent space of the Grassmannian at $[\boldsymbol{B}]$ is defined as $T_{\boldsymbol{B}} := \{\boldsymbol{W} \in \mathbb{R}^{D \times c} : \boldsymbol{W}^{\top}\boldsymbol{B} = \boldsymbol{0}\}$, and the orthogonal projector onto the tangent space is $\boldsymbol{I} - \boldsymbol{BB}^{\top}$, which is well-defined and does not depend on the class representative as $\boldsymbol{AA}^{\top} = \boldsymbol{BB}^{\top}$ for any $\boldsymbol{A} \in [\boldsymbol{B}]$. We generalize the definition of the Clarke subdifferential and denote by $\widetilde{\partial} f$ the Riemannian subdifferential of $f$ [2]:

$$\widetilde{\partial} f(\boldsymbol{B}) := \mathrm{conv} \left\{ \lim_{i \to \infty} (\boldsymbol{I} - \boldsymbol{BB}^{\top}) \nabla f(\boldsymbol{B}_i) : \boldsymbol{B}_i \to \boldsymbol{B}, f \text{ differentiable at } \boldsymbol{B}_i \in \mathbb{O}(c, D) \right\}. \quad (4)$$

We say that $\boldsymbol{B}$ is a critical point of (1) if and only if $\boldsymbol{0} \in \widetilde{\partial} f(\boldsymbol{B})$, which is a necessary condition for being a minimizer to (1).

## 3 Projected Riemannian Subgradient Method

In this section, we state our key Riemannian regularitity condition (RRC,§3.1), propose a projected Riemannian subgradient method (§3.2) based on RRC, and analyze its convergence properties (§3.3).

### 3.1 $(\alpha, \epsilon, \boldsymbol{B}^{\star})$-Riemannian Regularity Condition (RRC)

**Definition 1.** *We say that $f : \mathbb{R}^{D \times c} \to \mathbb{R}$ satisfies the $(\alpha, \epsilon, \boldsymbol{B}^{\star})$-Riemannian regularity condition (RRC)[2] for parameters $\{\alpha, \epsilon\} > 0$ and $\boldsymbol{B}^{\star} \in \mathbb{O}(c, D)$, if for every $\boldsymbol{B} \in \mathbb{O}(c, D)$ satisfying $\mathrm{dist}(\boldsymbol{B}, \boldsymbol{B}^{\star}) \leq \epsilon$, there exists a Riemannian subgradient $\mathcal{G}(\boldsymbol{B}) \in \widetilde{\partial} f(\boldsymbol{B})$ such that*

$$\langle \mathcal{P}_{\boldsymbol{B}^{\star}}(\boldsymbol{B}) - \boldsymbol{B}, -\mathcal{G}(\boldsymbol{B}) \rangle \geq \alpha \, \mathrm{dist}(\boldsymbol{B}, \boldsymbol{B}^{\star}). \quad (5)$$

Recently, a particular instance of Definition 1 was shown to hold [2] in the context of ODL (see §4.2). We note that $-\mathcal{G}(\boldsymbol{B})$ is not necessarily a descent direction for all $\mathcal{G}(\boldsymbol{B}) \in \widetilde{\partial} f(\boldsymbol{B})$, and that the set of allowable Riemannian subgradients that satisfy (5) need not include the minimum norm element from $\widetilde{\partial} f(\boldsymbol{B})$ even though that one is known to be a descent direction [18]. In §4, we show that a natural choice of Riemannian subgradient satisfies (5) for DPCP and ODL, where $\boldsymbol{B}^{\star}$ is a target solution. As illustrated in Figure 1, condition (5) implies that the negative of the chosen Riemannian subgradient $\mathcal{G}(\boldsymbol{B})$ has small angle to the direction $\mathcal{P}_{\boldsymbol{B}^{\star}}(\boldsymbol{B}) - \boldsymbol{B}$. To see this, let

$$\xi := \sup \{\|\mathcal{G}(\boldsymbol{B})\|_F : \mathrm{dist}(\boldsymbol{B}, \boldsymbol{B}^{\star}) \leq \epsilon\} \quad (6)$$

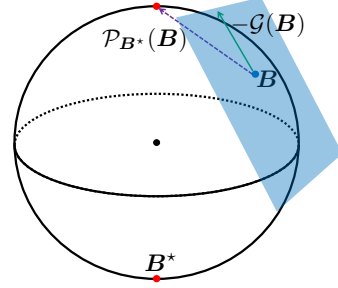

Figure 1: Illustration of Definition 1. Red nodes denote $[\boldsymbol{B}^{\star}]$, with the top one closest to $\boldsymbol{B}$. Inequality (5) requires the angle between $\mathcal{P}_{\boldsymbol{B}^{\star}}(\boldsymbol{B}) - \boldsymbol{B}$ (purple arrow) and $-\mathcal{G}(\boldsymbol{B})$ (blue arrow) to be sufficiently small.

denote an upper bound on the size of the Riemannian subgradients in a neighbrohood of $\boldsymbol{B}^{\star}$. Assume $\xi < \infty$. From (5) we have $\langle \mathcal{P}_{\boldsymbol{B}^{\star}}(\boldsymbol{B}) - \boldsymbol{B}, -\mathcal{G}(\boldsymbol{B}) \rangle / \|\mathcal{P}_{\boldsymbol{B}^{\star}}(\boldsymbol{B}) - \boldsymbol{B}\|_F \|\mathcal{G}(\boldsymbol{B})\|_F \geq \alpha/\xi$, which gives a bound on the sum of the cosines of the principal angles between $\mathcal{P}_{\boldsymbol{B}^{\star}}(\boldsymbol{B}) - \boldsymbol{B}$ and $-\mathcal{G}(\boldsymbol{B})$ and implies that $\xi \geq \alpha$.

In §3.3 we prove that if the $(\alpha, \epsilon, \boldsymbol{B}^{\star})$-RRC holds, then a projected Riemannian subgradient method will converge to $\boldsymbol{B}^{\star}$ when an appropriate initialization and step size strategy are used.

Definition 1 is similar in nature to other regularity conditions that characterize geometric properties of the objective function. Perhaps the most closely related ones for non-smooth functions are *sharpness* and *weak convexity*. Consider a function $h : \mathbb{R}^D \to \mathbb{R}$ and assume that the set

$$\mathcal{X} := \{\boldsymbol{z} \in \mathbb{R}^D : h(\boldsymbol{z}) \leq h(\boldsymbol{x}) \text{ for all } \boldsymbol{x} \in \mathbb{R}^n\}$$

of global minima of $h$ is non-empty. Then, $h$ is said to be sharp with parameter $\nu > 0$ (see [5]) if

$$h(\boldsymbol{x}) - \min_{\boldsymbol{z} \in \mathbb{R}^D} h(\boldsymbol{z}) \geq \nu \, \mathrm{dist}(\boldsymbol{x}, \mathcal{X}) \quad (7)$$

holds for all $\boldsymbol{x} \in \mathbb{R}^D$. The function $h$ is said to be weakly convex with parameter $\tau \geq 0$ if $\boldsymbol{x} \mapsto h(\boldsymbol{x}) + \frac{\tau}{2}\|\boldsymbol{x}\|^2$ is convex [44]. If $h$ is both sharp and weakly convex, then [10, 27] show that

$$\langle \mathcal{P}_{\mathcal{X}}(\boldsymbol{x}) - \boldsymbol{x}, \boldsymbol{d} \rangle \geq \nu \, \mathrm{dist}(\boldsymbol{x}, \mathcal{X}) - \frac{\tau}{2} \mathrm{dist}^2(\boldsymbol{x}, \mathcal{X}) \quad (8)$$

for any $\boldsymbol{x} \in \mathbb{R}^D$ and any $\boldsymbol{d} \in \partial h(\boldsymbol{x})$, where $\mathcal{P}_{\mathcal{X}}$ is the orthogonal projector onto the set $\mathcal{X}$. Note that (8) is useful when its right-hand side is nonnegative, i.e., when $\mathrm{dist}(\boldsymbol{x}, \mathcal{X}) \leq (2\nu)/\tau$. Thus, for any $\epsilon < (2\nu)/\tau$, we have

$$\langle \mathcal{P}_{\mathcal{X}}(\boldsymbol{x}) - \boldsymbol{x}, \boldsymbol{d} \rangle \geq \left(\nu - \frac{\tau}{2}\epsilon\right) \mathrm{dist}(\boldsymbol{x}, \mathcal{X}) \text{ for all } \boldsymbol{d} \in \partial h(\boldsymbol{x}) \quad (9)$$

whenever $\boldsymbol{x}$ satisfies $\mathrm{dist}(\boldsymbol{x}, \mathcal{X}) \leq \epsilon$. Noting the similarity between (9) and (5) ($\boldsymbol{B}^{\star}$ can be taken as a minimizer of $h$), the RRC (5) can be viewed as a generalization of (9) (the consequence of sharpness and weak convexity) to the Riemannian manifold. There are two main differences. First, (5) differs from (9) in that its left-hand side involves the Riemannian subgradient due to the Grassmannian constraint. Second, (5) is only required to hold for a particular Riemannian subgradient at $\boldsymbol{B}$, while (9) holds for all subgradients, thus imposing a slightly stronger regularity condition on the problem.

## 3.2  Projected Riemannian Subgradient Method on the Grassmannian

We propose to solve (1) using the projected Riemannian subgradient method in Algorithm 1. Given the $k$th iterate $\boldsymbol{B}_k$, the next iterate $\boldsymbol{B}_{k+1}$ is obtained by first moving in a direction opposite to a Riemannian subgradient at $\boldsymbol{B}_k$ that satisfies the regularity condition in (5), and then performing orthonormalization. In Section 4, we will show that such a projected Riemannian subgradient can be easily computed for ODL and DPCP. Note that $\widehat{\boldsymbol{B}}_{k+1}$ in (10) always has full column rank since $\mathcal{G}(\boldsymbol{B}_k)$ is orthogonal to $\boldsymbol{B}_k$; see the supplementary material for a formal proof. Also, there are multiple ways to orthonormalize $\widehat{\boldsymbol{B}}_{k+1}$, although for our purpose they are all equivalent since they all correspond to the same subspace. In (10), $\mathrm{orth}$ refers to any method that finds an orthonormal basis for $\mathrm{Span}(\widehat{\boldsymbol{B}}_{k+1})$. For example, one can compute $\boldsymbol{B}_{k+1}$ to be the Gram-Schmidt orthonormalization of $\widehat{\boldsymbol{B}}_{k+1}$, or as the first $c$ left singular vectors of $\widehat{\boldsymbol{B}}_{k+1}$. Finally, note that no specific step size rule is provided in Algorithm 1, whereas specific choices are made for the convergence analysis in §3.3.

---

**Algorithm 1** Projected Riemannian Subgradient Method

---

**Initialization:** set $\boldsymbol{B}_0$ and $\mu_0$;

1: **for** $k = 0, 1, \ldots$ **do**
2:     obtain $\mathcal{G}(\boldsymbol{B}_k) \in \widetilde{\partial} f(\boldsymbol{B}_k)$ satisfying (5) with $\boldsymbol{B} = \boldsymbol{B}_k$;
3:     compute a step size $\mu_k$ according to a certain rule;
4:     update the iterate:

$$\widehat{\boldsymbol{B}}_{k+1} \leftarrow \boldsymbol{B}_k - \mu_k \mathcal{G}(\boldsymbol{B}_k) \ \text{ and } \ \boldsymbol{B}_{k+1} \leftarrow \text{orth}(\widehat{\boldsymbol{B}}_{k+1}); \tag{10}$$

5: **end for**

---

### 3.3 Convergence Analysis

Our convergence analysis for Algorithm 1 relies in the RRC of Definition 1. When this regularity condition holds, we show that the iterates of Algorithm 1 exhibit the following properties: (i) they converge to a neighborhood of the set $\boldsymbol{B}^\star$ when a constant step size is used, and (ii) they converge at an R-linear rate to $\boldsymbol{B}^\star$ when a geometrically diminishing step size is used.

#### 3.3.1 Constant step size

We first consider the convergence of Algorithm 1 when a constant step size is used.

**Proposition 1.** *Suppose that for some* $(\alpha, \epsilon, \boldsymbol{B}^\star)$ *the function* $f$ *satisfies the* $(\alpha, \epsilon, \boldsymbol{B}^\star)$-*RRC in Definition 1. Let* $\{\boldsymbol{B}_k\}$ *be generated by Algorithm 1 with step size* $\mu_k \equiv \mu \leq \alpha\epsilon/\xi^2$ *and initial iterate* $\boldsymbol{B}_0$ *satisfying* $\text{dist}(\boldsymbol{B}_0, \boldsymbol{B}^\star) \leq \epsilon$, *where* $\xi$ *is defined in* (6). *Then, for all* $k \geq 0$, *it holds that*

$$\text{dist}(\boldsymbol{B}_k, \boldsymbol{B}^\star) \leq \max\left\{ \text{dist}(\boldsymbol{B}_0, \boldsymbol{B}^\star) - \mu\alpha k/2, \mu\xi^2/\alpha \right\}. \tag{11}$$

Towards interpreting Proposition 1, first consider the case $\text{dist}(\boldsymbol{B}_0, \boldsymbol{B}^\star) > \mu\xi^2/\alpha$, in which case (11) implies that after at most $K = 2(\text{dist}(\boldsymbol{B}_0, \boldsymbol{B}^\star) - \mu\xi^2/\alpha)/(\mu\alpha)$ iterates, the inequality $\text{dist}(\boldsymbol{B}_k, \boldsymbol{B}^\star) \leq \mu\xi^2/\alpha$ will hold for all $k \geq K$. In that sense, Proposition 1 essentially says that no further decay of $\text{dist}(\boldsymbol{B}_k, \boldsymbol{B}^\star)$ can be guaranteed. This agrees with empirical evidence regarding Algorithm 1 with constant step size (see Section 4). Note that (11) also suggests a tradeoff in selecting the step size $\mu$. A larger step size $\mu$ leads to a faster decrease on the bound but a larger universal upper bound of $\mu\xi^2/\alpha$, which may even exceed $\text{dist}(\boldsymbol{B}_0, \boldsymbol{B}^\star)$ if $\mu$ is too large.

#### 3.3.2 Geometrically diminishing step size

A useful strategy to balance the tradeoff discussed in the previous paragraph is to use a diminishing step size that starts relatively large and decreases to zero as the iterates proceed. As the universal upper bound $\frac{\mu\xi^2}{\alpha}$ in (11) is proportional to $\mu$, it is expected that the decay rate of the step size will determine the convergence rate of the iterates. In this section, we consider a geometrically diminishing step size scheme, i.e., we decrease the step size by a fixed fraction between iterations. Our argument is inspired by [10, 27]. Convergence with geometrically diminishing step size is guaranteed by the following result, which shows that if we choose the decay rate and initial step size properly, then the projected Riemannian subgradient method converges to $\boldsymbol{B}^\star$ at an R-linear rate.

**Theorem 1.** *Suppose that* $f$ *satisfies the* $(\alpha, \epsilon, \boldsymbol{B}^\star)$-*RRC in Definition 1. Let* $\{\boldsymbol{B}_k\}$ *be the sequence generated by Algorithm 1 with step size*

$$\mu_k = \mu_0 \beta^k \tag{12}$$

*and initialization* $\boldsymbol{B}_0$ *satisfying* $\text{dist}(\boldsymbol{B}_0, \boldsymbol{B}^\star) \leq \epsilon$. *Assume*

$$\mu_0 \leq \frac{\alpha\,\text{dist}(\boldsymbol{B}_0, \boldsymbol{B}^\star)}{2\xi^2} \quad \text{and} \quad \sqrt{1 - 2\frac{\alpha\mu_0}{\text{dist}(\boldsymbol{B}_0, \boldsymbol{B}^\star)} + \frac{\mu_0^2\xi^2}{\text{dist}^2(\boldsymbol{B}_0, \boldsymbol{B}^\star)}} =: \underline{\beta} \leq \beta < 1, \tag{13}$$

*where* $\xi$ *is defined in* (6). *Then, the sequence* $\{\boldsymbol{B}_k\}$ *satisfies*

$$\text{dist}(\boldsymbol{B}_k, \boldsymbol{B}^\star) \leq \text{dist}(\boldsymbol{B}_0, \boldsymbol{B}^\star)\beta^k \ \text{ for all } \ k \geq 0. \tag{14}$$

The rate at which $\{\operatorname{dist}(\boldsymbol{B}_k, \boldsymbol{B}^\star)\}_{k \geq 0}$ tends to zero in (14) is determined by $\beta$ which satisfies (13). Note that $\underline{\beta}$ is well defined and is strictly less than 1 in (13). To see this, on one hand, $\mu_0 \leq \alpha \operatorname{dist}(\boldsymbol{B}_0, \boldsymbol{B}^\star)/2\xi^2$ and $\xi \geq \alpha$ together imply $1 - 2\alpha\mu_0/\operatorname{dist}(\boldsymbol{B}_0, \boldsymbol{B}^\star) \geq 0$. On the other hand, $-2\alpha\mu_0/\operatorname{dist}(\boldsymbol{B}_0, \boldsymbol{B}^\star) + \mu_0^2\xi^2/\operatorname{dist}^2(\boldsymbol{B}_0, \boldsymbol{B}^\star) < 0$ is a decreasing function of $\mu_0$ when $\mu_0 \in (0, \alpha \operatorname{dist}(\boldsymbol{B}_0, \boldsymbol{B}^\star)/2\xi^2]$. In particular, when $\mu_0 = \alpha \operatorname{dist}(\boldsymbol{B}_0, \boldsymbol{B}^\star)/2\xi^2$, we have $\underline{\beta} = \sqrt{1 - 3\alpha^2/4\xi^2}$, giving the fastest decaying rate by setting $\beta = \underline{\beta}$. Finally, if $\operatorname{dist}(\boldsymbol{B}_0, \boldsymbol{B}^\star)$ is not known a priori, then one can replace it by its upper boud $\epsilon$ in (13) and (14) and the results still hold.

## 4   Applications

In this section, we show that Algorithm 1 achieves an R-linear convergence rate for Dual Principal Component Pursuit (DPCP) [42, 51] and Orthogonal Dictionary Learning (ODL) [39, 2].

### 4.1   DPCP for Robust Subspace Learning

We begin with the problem of learning a subspace from data corrupted by outliers [25]. Important methods include *Random Sampling And Consensus (RANSAC)* [17], fast median subspace [24], geodesic gradient descent [31], coherence pursuit [35], and many that solve convex formulations based on $\ell_1$ and nuclear norm optimization [46, 49, 26, 37, 48], but require either the dimension of the subspace or the number of outliers to be sufficiently small. On the other hand, DPCP [40, 41, 42, 51] solves a non-convex problem, can provably handle subspaces of high dimension, and can provably tolerate as many outliers as the square of the number of inliers. DPCP has been successfully applied in three-view geometry problems [42] and road plane detection from 3D point cloud data [51, 12], has been shown to outperform RANSAC, and has been applied in the multiple-hyperplane case [41]. The main principle behind DPCP is the computation of a basis for the orthogonal complement of the subspace to be learned. Specifically, given a dataset $\widetilde{\boldsymbol{\mathcal{X}}} = [\boldsymbol{\mathcal{X}} \, \boldsymbol{\mathcal{O}}]\boldsymbol{\Gamma} \in \mathbb{R}^{D \times L}$, where the columns of $\boldsymbol{\mathcal{X}} \in \mathbb{R}^{D \times N}$ are inlier points spanning a $d$-dimensional subspace $\mathcal{S}$ of $\mathbb{R}^D$, the columns of $\boldsymbol{\mathcal{O}} \in \mathbb{R}^{D \times M}$ are outlier points, and $\boldsymbol{\Gamma}$ is an unknown permutation, DPCP solves

$$\underset{\boldsymbol{B} \in \mathbb{O}(c,D)}{\operatorname{minimize}} f(\boldsymbol{B}) := \|\widetilde{\boldsymbol{\mathcal{X}}}^\top \boldsymbol{B}\|_{1,2} \equiv \sum_{i=1}^L \|\widetilde{\boldsymbol{x}}_i^\top \boldsymbol{B}\|_2, \tag{15}$$

where $c = D - d$ is the codimension of $\mathcal{S}$. An iterative reweighted least squares (IRLS) algorithm has been empirically utilized to solve (15) in [42], but without formal guarantees.

**Verification of the regularity condition.**    We will show that the DPCP problem (15) satisfies the RRC, which will then be used to establish convergence rates. Since the objective function $f$ in (15) is regular, it follows from [47] that $\widetilde{\partial} f(\boldsymbol{B}) = (\mathbf{I} - \boldsymbol{B}\boldsymbol{B}^\top)\partial f(\boldsymbol{B})$. Also note that the $\ell_2$ norm is subdifferentially regular, thus by the chain rule one choice for the Riemannian subgradient is

$$\mathcal{G}(\boldsymbol{B}) = (\mathbf{I} - \boldsymbol{B}\boldsymbol{B}^\top) \sum_{i=1}^L \widetilde{\boldsymbol{x}}_i \operatorname{sign}(\widetilde{\boldsymbol{x}}_i^\top \boldsymbol{B}), \text{ where } \operatorname{sign}(\boldsymbol{a}) := \begin{cases} \boldsymbol{a}/\|\boldsymbol{a}\|_2 & \text{if } \boldsymbol{a} \neq \boldsymbol{0}, \\ \boldsymbol{0} & \text{if } \boldsymbol{a} = \boldsymbol{0}. \end{cases} \tag{16}$$

To analyze (15), we define two quantities. The first one characterizes the maximum Riemannian subgradient related to outliers: $\eta_{\boldsymbol{\mathcal{O}}} := \frac{1}{M} \max_{\boldsymbol{B} \in \mathbb{O}(c,D)} \left\| (\mathbf{I} - \boldsymbol{B}\boldsymbol{B}^\top) \sum_{i=1}^M \boldsymbol{o}_i \operatorname{sign}(\boldsymbol{o}_i^\top \boldsymbol{B}) \right\|_F$, which appears in [51] when $\boldsymbol{B}$ is on $\mathbb{O}(1, D)$. The second one is related to the inliers and is given by $c_{\boldsymbol{\mathcal{X}},\min} := \frac{1}{N} \min_{\boldsymbol{b} \in \mathcal{S} \cap \mathbb{O}(1,D)} \|\boldsymbol{\mathcal{X}}^\top \boldsymbol{b}\|_1$, which is referred to as the *permeance statistic* in [26]. These quantities reflect how well distributed the inliers and outliers are, with larger values of $c_{\boldsymbol{\mathcal{X}},\min}$ (respectively, smaller values of $\eta_{\boldsymbol{\mathcal{O}}}$) corresponding to a more uniform distributions of inliers (respectively, outliers). One of the key insights in this paper is that the DPCP problem (15) satisfies the RRC of Definition 1 as we now state.

**Theorem 2.** *For any* $\epsilon < \sqrt{2\left(1 - M\eta_{\boldsymbol{\mathcal{O}}}/Nc_{\boldsymbol{\mathcal{X}},\min}\right)}$, *the DPCP problem* (15) *satisfies the* $(\alpha, \epsilon, \boldsymbol{S}^\perp)$-*RRC with* $\alpha = ((1 - \epsilon^2/2)Nc_{\boldsymbol{\mathcal{X}},\min} - M\eta_{\boldsymbol{\mathcal{O}}})/\sqrt{2c}$ *and any orthonormal basis* $\boldsymbol{S}^\perp$ *for* $\mathcal{S}^\perp$. *Also,* $\|\mathcal{G}(\boldsymbol{B})\|_F \leq \sqrt{N} \|\boldsymbol{\mathcal{X}}\|_2 + M\eta_{\boldsymbol{\mathcal{O}}}$ *for all* $\boldsymbol{B} \in \mathbb{O}(c, D)$, *where* $\|\cdot\|_2$ *denotes the spectral norm.*

Combining this with Theorem 1 allows us to conclude the linear convergence of Algorithm 1 to $\boldsymbol{S}^\perp$.

**Corollary 1.** *Suppose that the initialization $\boldsymbol{B}_0$ satisfies $\mathrm{dist}^2(\boldsymbol{B}_0, \boldsymbol{S}^\perp) < 2\left(1 - M\eta_{\mathcal{O}}/Nc_{\boldsymbol{\mathcal{X}},\min}\right)$, where $\mathrm{dist}(\boldsymbol{B}_0, \boldsymbol{S}^\perp)$ is defined in (2). Let $\{\boldsymbol{B}_k\}$ be the sequence generated by Algorithm 1 for solving the DPCP problem (15) with $\mathcal{G}(\boldsymbol{B}_k)$ in (16) and step size $\mu_k = \mu_0\beta^k$, where $\mu_0$ and $\beta$ satisfy (13) with $\epsilon = \mathrm{dist}(\boldsymbol{B}_0, \boldsymbol{S}^\perp)$, $\alpha = ((1 - \epsilon^2/2)Nc_{\boldsymbol{\mathcal{X}},\min} - M\eta_{\mathcal{O}})/\sqrt{2}c$, and $\xi = \sqrt{N}\|\boldsymbol{\mathcal{X}}\|_2 + M\eta_{\mathcal{O}}$. Then, $\boldsymbol{B}_k$ converges to $\boldsymbol{S}^\perp$ at an R-linear rate, i.e., $\mathrm{dist}(\boldsymbol{B}_k, \boldsymbol{S}^\perp) \leq \beta^k \mathrm{dist}(\boldsymbol{B}_0, \boldsymbol{S}^\perp)$ for all $k \geq 0$.*

Corollary 1 implies that the Riemannian subgradient method with a good initialization converges to an orthonormal basis of $\boldsymbol{S}^\perp$ at an R-linear rate. When $c = 1$, a projected subgradient method was proved to have a piecewise linear convergence rate in [51]. In this case, Corollary 1 improves upon [51] in three ways: (i) it allows for a simpler strategy for selecting the step size than does the piecewise geometrically diminishing step size, which has two more parameters controlling when and how often to decay the step size; (ii) it provides a more transparent convergence analysis since its proof follows directly from the RRC and Theorem 1; and (iii) it places a slightly weaker requirement on the initialization, which in practice we compute as the bottom eigenvectors of $\widetilde{\boldsymbol{\mathcal{X}}}\widetilde{\boldsymbol{\mathcal{X}}}^\top$ as in [42, 51]. In the supplementary material, we show this spectral initialization satisfies the requirement in Corollary 1.

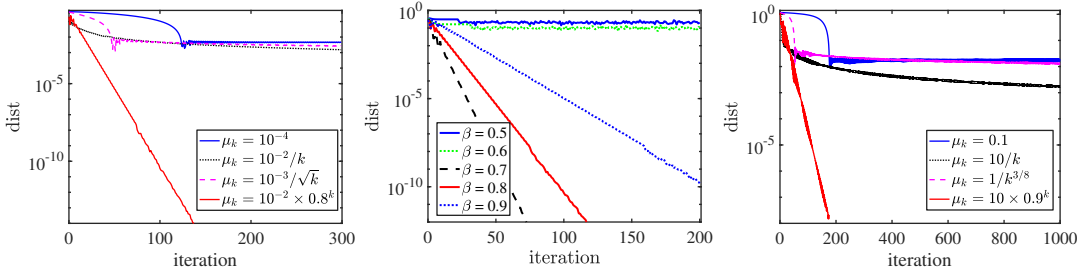

(a) Performance on problem (15) with different step size choices $\mu_k$.  (b) Performance on problem (15) for the step size $\mu_k = 0.01\beta^k$.  (c) Performance on problem (17) with different step sizes choices $\mu_k$.

Figure 2: Performance of Algorithm 1 on the DPCP problem (15) and the ODL problem (17).

**Experiments.** Synthetic data for the DPCP problem is generated as follows: randomly sample a subspace $\mathcal{S}$ of co-dimension $c = 10$ in ambient dimension $D = 100$, and uniformly at random sample $N = 1500$ inliers from $\mathcal{S} \cap \mathbb{O}(1, D)$ and $M = 3500$ outliers from $\mathbb{O}(1, D)$ so that the outlier ratio is $M/(M + N) = 0.7$. As an initialization $\boldsymbol{B}_0$, we use the bottom $c$ eigenvectors of $\widetilde{\boldsymbol{\mathcal{X}}}\widetilde{\boldsymbol{\mathcal{X}}}^\top$ [42, 51] as we described before.

Figure 2a displays the convergence of the projected Riemannian subgradient method with different choices of step size. We observe linear convergence for the geometrically diminishing step size, which converges much faster than when a constant step size or classical diminishing step size ($O(1/k)$ and $O(1/\sqrt{k})$) is used. In Figure 2b, we illustrate the effect of the decay factor $\beta$ for Algorithm 1 with geometrically diminishing step size $\mu_k = 0.01\beta^k$. First observe that, as expected, $\beta$ controls the convergence speed. When $\beta$ is too small (e.g., $\beta \in \{0.5, 0.6\}$) convergence may not occur, which agrees with (13) and (14). However, when $\beta \geq 0.7$ the algorithm converges at an R-linear rate, with larger values of $\beta$ resulting in slower convergence speeds.

## 4.2 Orthogonal Dictionary Learning

Given a dataset $\widetilde{\boldsymbol{\mathcal{X}}} \in \mathbb{R}^{D \times N}$, DL [32] aims to learn a sparse representation $\boldsymbol{\Theta} \in \mathbb{R}^{M \times N}$ for $\widetilde{\boldsymbol{\mathcal{X}}}$ by finding a dictionary $\boldsymbol{A} \in \mathbb{R}^{D \times M}$ such that $\widetilde{\boldsymbol{\mathcal{X}}} \approx \boldsymbol{A}\boldsymbol{\Theta}$ with $\boldsymbol{\Theta}$ sparse. Several DL methods have been proposed in the literature, including the method of optimal directions (MOD) [16], K-SVD [15], and alternating minimization [1], as well as the Riemannian trust region method [39] and projected Riemannian subgradient method [2] for ODL. Here, we consider the ODL problem [39, 2] in which the dictionary is square and orthogonal and the data is generated by the following random model.[3]

**Definition 2** (Random model for ODL [2]). *Assume $\boldsymbol{A} \in \mathbb{R}^{D \times D}$ is a fixed but unknown orthonormal matrix. The data is generated as $\widetilde{\boldsymbol{\mathcal{X}}} = \boldsymbol{A}\boldsymbol{\Theta}$, where each column of $\boldsymbol{\Theta} \in \mathbb{R}^{D \times D}$ is an i.i.d. Bernoulli-Gaussian random vector with parameter $\rho \in (0, 1)$ that controls the sparsity.*

If $\boldsymbol{b}$ is a column of $\boldsymbol{A}$, then $\boldsymbol{A}^\top \boldsymbol{b}$ is a standard basis vector and $\widetilde{\boldsymbol{\mathcal{X}}}^\top \boldsymbol{b} = \boldsymbol{\Theta}^\top \boldsymbol{A}^\top \boldsymbol{b}$ is sparse. Thus, minimizing $\|\widetilde{\boldsymbol{\mathcal{X}}}^\top \boldsymbol{b}\|_0$ over the sphere is expected to yield the column of $\boldsymbol{A}$ that is least used in the representation $\boldsymbol{\Theta}$. For computational reasons, the $\ell_0$ semi-norm is replaced by the $\ell_1$ norm [2][4] thus leading to the problem[5]

$$\underset{\boldsymbol{b} \in \mathbb{O}(1,D)}{\text{minimize}} \, f(\boldsymbol{b}) = \tfrac{1}{N}\|\widetilde{\boldsymbol{\mathcal{X}}}^\top \boldsymbol{b}\|_1. \tag{17}$$

**Verification of the regularity condition.** We show that the regularity condition in (5) is satisfied for problem (17). The primary difference with our previous analysis is that the data is now random. Thus, (5) will only be proved to hold with high probability. Towards that end, similar to (16), a Riemannian subgradient for (17) is

$$\mathcal{G}(\boldsymbol{b}) = \tfrac{1}{N}\big(\mathbf{I} - \boldsymbol{b}\boldsymbol{b}^\top\big)\Big( \sum_{i=1}^{N} \text{sign}(\widetilde{\boldsymbol{x}}_i^\top \boldsymbol{b})\widetilde{\boldsymbol{x}}_i \Big). \tag{18}$$

The projected Riemannian subgradient method has been utilized in [2] for solving (17), but only with a sublinear rate of convergence guarantee, even though the function has been proved to satisfy (5) with high probability. Based on this condition, we will show that Algorithm 1 can solve (17) more efficiently, indeed with a linear convergence rate. To describe the RRC for (17), suppose without loss of generality that the orthonormal dictionary $\boldsymbol{A}$ is the identity matrix. Then, the goal is to find the standard basis vectors $\{\pm\boldsymbol{e}_1, \ldots, \pm\boldsymbol{e}_D\}$, where the sign is irrelevant because $f(\boldsymbol{b}) = f(-\boldsymbol{b})$. We now define a region of interest that is near each basis vector $\boldsymbol{e}_i$ and $-\boldsymbol{e}_i$ as $\mathcal{I}_\zeta^i = \left\{ \boldsymbol{b} \in \mathbb{O}(1,D) : \frac{b_i^2}{\max_{j \neq i} b_j^2} \geq 1 + \zeta \right\}$, where $\zeta > 0$ and $b_j$ is the $j$-th entry of $\boldsymbol{b}$. Each region $\mathcal{I}_\zeta^i$ contains all unit vectors whose $i$-th entry is at least $\sqrt{1+\zeta}$ larger (in absolute value) than the other entries. The RRC for (17) is then captured by the following result.

**Theorem 3.** *[2, Theorem 3.6] Assume $\rho \in [1/D, 1/2]$ in the random model of Definition 2. There exist universal constants $C, c > 0$ such that if $N \geq CD^4\zeta^{-2}\rho^{-2}\log(D/\zeta)$ for all $\zeta \in (0,1)$, then with probability at least $1 - \exp(-cN\rho^3\zeta^2 D^{-3}/\log N)$ the ODL problem (17) satisfies (5) for any $\boldsymbol{b} \in \mathcal{I}_\zeta^i$ with $\mathcal{G}(\boldsymbol{b})$ in (18) and $\boldsymbol{B}^\star = \boldsymbol{e}_i$ for any $i$, and $\alpha = \frac{1}{16}\rho(1-\rho)\zeta D^{-\frac{3}{2}}$.*

Note that Theorem 3 ensures that (5) holds only for all $\boldsymbol{b} \in \mathcal{I}_\zeta^i$, but not all $\boldsymbol{b}$ that is $\epsilon$-close to $\boldsymbol{e}_i$. Fortunately, [2, Proposition D.2] ensures the iterates generated by Algorithm 1 do stay within $\mathcal{I}_\zeta^i$, which together with Theorem 1 guarantees the convergence of Algorithm 1.

**Corollary 2.** *Let $\{\boldsymbol{b}_k\}$ be the sequence generated by Algorithm 1 for the ODL problem (17) with $\boldsymbol{b}_0 \in \mathcal{I}_\zeta^i$ ($\zeta \leq \frac{55}{64}$) and step size $\mu_k = \mu_0\beta^k$, where $\mu_0$ and $\beta$ satisfy the conditions in Theorem 1 with $\xi = 2$ and $\epsilon = \sqrt{2}$, and $\alpha = \frac{1}{16}\rho(1-\rho)\zeta D^{-\frac{3}{2}}$. Under the same setup as in Theorem 3, with probability at least $1 - \exp(-cN\rho^3\zeta^2 D^{-3}/\log N)$, $\{\boldsymbol{b}_k\}$ converges to $\boldsymbol{e}_i$ at an R-linear rate, i.e.,*

$$\text{dist}(\boldsymbol{b}_k, \boldsymbol{e}_i) \leq \beta^k \text{dist}(\boldsymbol{b}_0, \boldsymbol{e}_i). \tag{19}$$

Corollary 2 improves upon [2, Theorem 3.8], according to which under the setup in Corollary 2 and with step size $\mu_k = O(1/k^{3/8})$, it follows that $\min_{k' \leq k} \text{dist}(\boldsymbol{b}_{k'}, \boldsymbol{e}_i) = O(1/k^{3/8})$. Indeed, our result (19) gives a direct bound on the $k$-th iteration and not on the best iteration obtained so far.

**Experiments** We use the same setup in [2] by first generating a random orthogonal dictionary $\boldsymbol{A} \in \mathbb{R}^{D \times D}$ with $D = 70$, sparsity level $\rho = 0.3$, and number of data points $N = 5857 \approx 10D^{1.5}$. As in [2], the initialization $\boldsymbol{b}_0$ is randomly generated from the unit sphere $\mathbb{O}(1, D)$ and belongs to one of the $D$ sets $\{\mathcal{I}_{1/5\log D}^i : i = 1, \ldots, D\}$ with probability at least one-half [2, Lemma 3.9]. Figure 2c

displays the convergence of the Riemannian subgradient method for different choices of the step size for solving (17). We observe linear convergence for geometrically diminishing step size, which converges much faster than the others, in particular when $\mu_k = O(\frac{1}{k^{3/8}})$ as is used in [2].

## 5  Conclusion and Discussion

We proved that a projected Riemannian subgradient method with geometrically diminishing step sizes converges linearly for non-convex and non-smooth problems on the Grassmannian that satisfy a certain regularity condition on the Riemannian subgradient. We also showed that our regularity condition is satisfied by $\ell_1$ co-sparse formulations for orthogonal dictionary learning and robust subspace learning, which led to improved convergence rates when compared to existing results.

We conclude this paper by pointing out several interesting directions for future work.

**Extension to intrinsic methods.** In this paper we take an extrinsic approach because extrinsic methods are typically easier to implement, e.g. when the projection map is easier to compute than the geodesic distance. Extending the current analysis to an intrinsic optimization method—where the iterates are taken along a geodesic direction—is worth exploring. For example, the geodesic gradient descent (GGD) [31] has been proved to converge at a piecewise linear rate for the robust subspace learning problem. And extension of the current analysis may allow the GGD to use a simpler step size selection strategy (i.e., a geometrically diminishing step size) to obtain a linear convergence rate.

**Extension to other submanifolds of Euclidean space.** Although we focus on optimization problems over the Grassmannian, the Riemannian regularity condition can be extended to other submanifolds of Euclidean space with an appropriate definition of the Riemanniann metric and distance. Using this condition to analyze the convergence of the projected Riemannian subgradient method for other manifolds (such as the Stiefel manifold) is the subject of ongoing work.

**Application to other problems.** Aside from the robust subspace and dictionary learning problems considered here, other problems in machine learning and signal processing can be formulated as minimizing a non-smooth function over the sphere or Stiefel manifold and thus (potentially) can be efficiently solved by the Riemannian subgradient method. These problems include the $\ell_1$-norm (kernel) PCA [22, 30, 45], multi-channel sparse blind deconvolution [28, 33], etc.

## Acknowledgment

This research is supported in part by NSF grant 1704458, ShanghaiTech grant 2017F0203-000-16, and the Northrop Grumman Mission Systems Research in Applications for Learning Machines (REALM) initiative. Zhihui Zhu would like to thank Xiao Li and Dr. Anthony Man-Cho So (CUHK) for fruitful discussions about regularity conditions for non-smooth optimization problems.

## Footnotes

[1] Suppose the sequence $\{\boldsymbol{x}_k\}$ converges to $\boldsymbol{x}^\star$. We say it converges sublinearly if $\lim_{k \to \infty} \|\boldsymbol{x}_{k+1} - \boldsymbol{x}^\star\| / \|\boldsymbol{x}_k - \boldsymbol{x}^\star\| = 1$, and R-linearly if there exists $C > 0, q \in (0, 1)$ such that $\|\boldsymbol{x}_k - \boldsymbol{x}^\star\| \leq Cq^k, \ \forall k \geq 0$.

[2]Strictly speaking, Definition 1 is extrinsic since we view the Grassmannian as embedded in the Euclidean space and (5) involves the standard inner product in the Euclidean space.

[3]Extensions to other models including deterministic models are the subject of future work.

[4][39] considered a smoothed version of (17), allowing one to use gradient-based algorithms. However, the obtained solution is perturbed from the targeted one and thus a rounding step is needed.

[5]All the columns of $\boldsymbol{A}$ can be obtained by repeating this process with the removal of the contribution from the previously learned columns. Alternatively, as will been seen in Corollary 2, the column that Algorithm 1 converges to depends on the initialization. Thus, one may simply repeat Corollary 2 with different initializations (e.g., random initializations) each time [2]. It is of interest to extend (17) in order to estimate the whole dictionary, e.g., minimizing $\|\widetilde{\boldsymbol{\mathcal{X}}}^\top \boldsymbol{B}\|_1$, s. t. $\boldsymbol{B} \in \mathbb{O}(D)$, where $\ell_1$ counts the sum of the elements of the matrix. Note that this is not an optimization on the Grassmannian since the objective is not rotation invariant.

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
