[Supplementary Material]

Supplementary Material for:
"A Linearly Convergent Method for Non-smooth Non-convex
Optimization on the Grassmannian with Applications to
Robust Subspace and Dictionary Learning"

May 23, 2019

## Contents

# 1    Basic Notations and Definitions

We consider the following problem [1]

$$\underset{\boldsymbol{B}\in\mathbb{O}(c,D)}{\text{minimize}} f(\boldsymbol{B}) \tag{1}$$

where $f : \mathbb{R}^{D\times c} \to \mathbb{R}$ is lower semi-continuous, possibly non-convex and non-smooth, and homogenous that $f(\boldsymbol{B}) = f(\boldsymbol{BQ})$ for any $\boldsymbol{Q} \in \mathbb{O}(c)$ . Note that the global minimum of (1) is not unique as if $\boldsymbol{B}^\star$ is a global minimum, then any point in $[\boldsymbol{B}^\star]$ is also a global minimum.

For any $\boldsymbol{A}, \boldsymbol{B} \in \mathbb{O}(c, D)$, the principal angles between $\text{Span}(\boldsymbol{A})$ and $\text{Span}(\boldsymbol{B})$ are defined as [2] $\phi_i(\boldsymbol{A}, \boldsymbol{B}) = \arccos(\sigma_i(\boldsymbol{A}^\top \boldsymbol{B})), i = 1, \ldots, c$, where $\sigma_i(\cdot)$ denotes the $i$-th singular value. We then define the distance between $\boldsymbol{A}$ and $\boldsymbol{B}$ as

$$\text{dist}(\boldsymbol{A}, \boldsymbol{B}) := \sqrt{2\sum_{i=1}^{c} \big(1 - \cos(\phi_i(\boldsymbol{A}, \boldsymbol{B}))\big)} = \inf_{\boldsymbol{Q}\in\mathbb{O}(c)} \|\boldsymbol{B} - \boldsymbol{AQ}\|_F, \tag{2}$$

where the last term is also known as the *orthogonal Procrustes problem* and the last equality follows from the result [3] that the optimal rotation matrix $\boldsymbol{Q}$ minimizing $\|\boldsymbol{B} - \boldsymbol{AQ}\|_F$ is equal to $\boldsymbol{Q} = \boldsymbol{UV}^\top$, where $\boldsymbol{U\Sigma V}^\top$ is the SVD of $\boldsymbol{A}^\top \boldsymbol{B}$. We also define the projection of $\boldsymbol{B}$ onto $[\boldsymbol{A}]$ as

$$\mathcal{P}_{\boldsymbol{A}}(\boldsymbol{B}) = \boldsymbol{AQ}^\star, \quad \text{where} \quad \boldsymbol{Q}^\star = \underset{\boldsymbol{Q}\in\mathbb{O}(c)}{\arg\min} \|\boldsymbol{B} - \boldsymbol{AQ}\|_F. \tag{3}$$

Since $f$ can be non-smooth and non-convex, we utilize the Fréchet subdifferential, which generalizes the gradient for smooth functions and the subdifferential in convex analysis. The subdifferential of a lower semi-continuous function $f$ at $\boldsymbol{B}$ is defined as

$$\partial f(\boldsymbol{B}) := \left\{ \boldsymbol{D} \in \mathbb{R}^{D\times c} : \liminf_{\boldsymbol{A}\to\boldsymbol{B}} \frac{f(\boldsymbol{A}) - f(\boldsymbol{B}) - \langle \boldsymbol{D}, \boldsymbol{A} - \boldsymbol{B}\rangle}{\|\boldsymbol{B} - \boldsymbol{A}\|_F} \geq 0 \right\}.$$

Roughly speaking, for each subgradient of $f$ at $\boldsymbol{B}$ (i.e., for each $\boldsymbol{D} \in \partial f(\boldsymbol{B})$), the graph of $\boldsymbol{A} \mapsto f(\boldsymbol{B}) + \langle \boldsymbol{D}, \boldsymbol{A} - \boldsymbol{B}\rangle$ constructs a local supporting hyperplane to the graph of $f$ at $\boldsymbol{B}$. If $f$ is a convex function, then a local supporting hyperplane turns out to be a global one, and $\partial f(\boldsymbol{B})$ reduces to $\big\{\boldsymbol{D} \in \mathbb{R}^{D\times c} : f(\boldsymbol{A}) - f(\boldsymbol{B}) \geq \langle \boldsymbol{D}, \boldsymbol{A} - \boldsymbol{B}\rangle$ for all $\boldsymbol{A} \in \mathbb{R}^{D\times c}\big\}$ [4]. If $f$ is a smooth function, then the subdifferential $\partial f(\boldsymbol{B})$ is simply $\{\nabla f(\boldsymbol{B})\}$.

Since we consider problems on the Grassmannian, we use tools from Riemannian geometry to state optimality conditions. From [1], the tangent space of the Grassmannian at $[\boldsymbol{B}]$ is defined as $\{\boldsymbol{W} \in \mathbb{R}^{D\times c} : \boldsymbol{W}^\top \boldsymbol{B} = \boldsymbol{0}\}$, and the orthogonal projector onto the tangent space is $\mathbf{I} - \boldsymbol{BB}^\top$, which is well-defined and does not depend on the class representative as $\boldsymbol{AA}^\top = \boldsymbol{BB}^\top$ for any $\boldsymbol{A} \in [\boldsymbol{B}]$. If $\boldsymbol{D} \in \partial f(\boldsymbol{B})$, then we call $(\mathbf{I} - \boldsymbol{BB}^\top)\boldsymbol{D}$ a Riemannian subgradient of $f$ at $\boldsymbol{B}$; we define the collection of all such Riemannian subgradients of $f$ at $\boldsymbol{B}$ as

$$\widetilde{\partial} f(\boldsymbol{B}) := \left\{ (\mathbf{I} - \boldsymbol{BB}^\top)\boldsymbol{D} : \boldsymbol{D} \in \partial f(\boldsymbol{B}) \right\}. \tag{4}$$

We say that $\boldsymbol{B}$ is a critical point of (1) if $\boldsymbol{0} \in \widetilde{\partial} f(\boldsymbol{B})$, which is a necessary condition for being a minimizer to (1) as shown in [5].

Armed with the Riemannian subgradient, a simple yet popular method for solving (1) is the projected Riemannian subgradient method, which is stated in Algorithm 1.

---
**Algorithm 1** Projected Riemannian Subgradient Method
---
**Initialization:** set $\boldsymbol{B}_0$ and $\mu_0$;

 1: **for** $k = 0, 1, \ldots$ **do**
 2:     obtain $\mathcal{G}(\boldsymbol{B}_k) \in \widetilde{\partial} f(\boldsymbol{B}_k)$ satisfying (6) with $\boldsymbol{B} = \boldsymbol{B}_k$;
 3:     compute a step size $\mu_k$ according to a certain rule;
 4:     update the iterate:

$$\widehat{\boldsymbol{B}}_{k+1} \leftarrow \boldsymbol{B}_k - \mu_k \mathcal{G}(\boldsymbol{B}_k), \quad \boldsymbol{B}_{k+1} \leftarrow \mathrm{orth}(\widehat{\boldsymbol{B}}_{k+1}); \tag{5}$$

 5: **end for**
---

## 1.1   $(\alpha, \epsilon, \boldsymbol{B}^\star)$-Riemannian Regularity Condition (RRC)

**Definition 1.** *We say that $f : \mathbb{R}^{D \times c} \to \mathbb{R}$ satisfies the $(\alpha, \epsilon, \boldsymbol{B}^\star)$-Riemannian regularity condition (RRC) for parameters $\{\alpha, \epsilon\} > 0$ and $\boldsymbol{B}^\star \in \mathbb{O}(c, D)$, if for every $\boldsymbol{B} \in \mathbb{O}(c, D)$ satisfying $\mathrm{dist}(\boldsymbol{B}, \boldsymbol{B}^\star) \leq \epsilon$, there exists a Riemannian subgradient $\mathcal{G}(\boldsymbol{B}) \in \widetilde{\partial} f(\boldsymbol{B})$ such that*

$$\langle \mathcal{P}_{\boldsymbol{B}^\star}(\boldsymbol{B}) - \boldsymbol{B}, -\mathcal{G}(\boldsymbol{B}) \rangle \geq \alpha \, \mathrm{dist}(\boldsymbol{B}, \boldsymbol{B}^\star). \tag{6}$$

Let

$$\xi := \sup \left\{ \|\mathcal{G}(\boldsymbol{B})\|_F : \mathrm{dist}(\boldsymbol{B}, \boldsymbol{B}^\star) \leq \epsilon \right\} \tag{7}$$

denote an upper bound on the size of the Riemannian subgradients in a neighborhood of $\boldsymbol{B}^\star$. Assume $\xi < \infty$. To compare $\alpha$ and $\xi$, we plug the Cauchy-Schwarz inequality $\langle \boldsymbol{B} - \mathcal{P}_{\boldsymbol{B}^\star}(\boldsymbol{B}), \mathcal{G}(\boldsymbol{B}) \rangle \leq \|\mathcal{G}(\boldsymbol{B})\|_F \, \mathrm{dist}(\boldsymbol{B}, \boldsymbol{B}^\star)$ into (6), giving

$$\|\mathcal{G}(\boldsymbol{B})\|_F \geq \alpha, \ \forall \ \boldsymbol{B} \notin [\boldsymbol{B}^\star], \mathrm{dist}(\boldsymbol{B}, \boldsymbol{B}^\star) \leq \epsilon, \tag{8}$$

which implies that

$$\xi \geq \alpha. \tag{9}$$

## 2   Proof of Proposition 1

We first repeat Proposition 1.

**Proposition 1.** *Suppose that for some $(\alpha, \epsilon, \boldsymbol{B}^\star)$ the function $f$ satisfies the $(\alpha, \epsilon, \boldsymbol{B}^\star)$-RRC in Definition 1. Let $\{\boldsymbol{B}_k\}$ be generated by Algorithm 1 with step size $\mu_k \equiv \mu \leq \alpha \epsilon / \xi^2$ and initial iterate $\boldsymbol{B}_0$ satisfying $\mathrm{dist}_0 \leq \epsilon$. Then, for all $k \geq 0$, it holds that*

$$\mathrm{dist}(\boldsymbol{B}_k, \boldsymbol{B}^\star) \leq \max \left\{ \mathrm{dist}(\boldsymbol{B}_0, \boldsymbol{B}^\star) - \mu \alpha k / 2, \mu \xi^2 / \alpha \right\}. \tag{10}$$

*Proof of Proposition 1.* We first prove that $\widehat{\boldsymbol{B}}_{k+1}$ always has full column rank since $\mathcal{G}(\boldsymbol{B}_k)$ is orthogonal to $\boldsymbol{B}_k$. Let $\widehat{\boldsymbol{B}}_{k+1} = \boldsymbol{P} \boldsymbol{\Omega} \boldsymbol{Q}^\top$ be a reduced SVD of $\widehat{\boldsymbol{B}}_{k+1}$, where $\boldsymbol{\Omega}$ is an $c \times c$ diagonal matrix with singular values $w_1, \ldots, w_c$ along the diagonals. Since $\widehat{\boldsymbol{B}}_{k+1} = \boldsymbol{B}_k - \mu_k \mathcal{G}(\boldsymbol{B}_k)$, we have

$$\widehat{\boldsymbol{B}}_{k+1}^\top \widehat{\boldsymbol{B}}_{k+1} = \mathbf{I} + \mu_k^2 \left( \mathcal{G}(\boldsymbol{B}_k) \right)^\top \mathcal{G}(\boldsymbol{B}_k),$$

where the equality follows because $\boldsymbol{B}_k \in \mathbb{O}(c, D)$ is orthogonal to $\mathcal{G}(\boldsymbol{B}_k)$. Thus, the eigenvalues of $\widehat{\boldsymbol{B}}_{k+1}^{\top}\widehat{\boldsymbol{B}}_{k+1}$ is always greater than or equal to 1, which implies that $w_1, \ldots, w_c \geq 1$. Therefore, all singular values of $\widehat{\boldsymbol{B}}_{k+1}$ are non-vanishing, which means $\widehat{\boldsymbol{B}}_{k+1}$ has full column rank. Additionally, for any $\boldsymbol{U} \in \mathbb{O}(c, D)$, it follows that

$$
\begin{aligned}
& \|\widehat{\boldsymbol{B}}_{k+1} - \boldsymbol{U}\|_F^2 - \|\boldsymbol{B}_{k+1} - \boldsymbol{U}\|_F^2 \\
&= \|\boldsymbol{P}\boldsymbol{\Omega}\boldsymbol{Q}^{\top}\|_F^2 - \|\boldsymbol{P}\boldsymbol{Q}^{\top}\|_F^2 - 2\operatorname{trace}((\boldsymbol{\Omega} - \mathbf{I})\boldsymbol{P}^{\top}\boldsymbol{U}\boldsymbol{Q}) \\
&\geq \sum_{i=1}^{c} \omega_i^2 - 1 - 2(\omega_i - 1) = \sum_{i=1}^{c}(\omega_i - 1)^2 \geq 0,
\end{aligned}
\tag{11}
$$

where we have chosen $\boldsymbol{B}_{k+1}$ to be $\boldsymbol{P}\boldsymbol{Q}^{\top}$, and the last line directly follows Von Neumann's inequality, i.e., $\operatorname{trace}(\boldsymbol{F}^{\top}\boldsymbol{G}) \leq \sum_i \sigma_i(\boldsymbol{F})\sigma_i(\boldsymbol{G})$ where $\sigma_1(\cdot) \geq \sigma_2(\cdot) \geq \cdots \geq 0$.

We now prove (10) by induction. It is clear that (10) holds when $k = 0$. Now assume that (10) holds at the $k$-th iteration, which implies that $\operatorname{dist}(\boldsymbol{B}_k, \boldsymbol{B}^{\star}) \leq \epsilon$. Then,

$$
\begin{aligned}
\operatorname{dist}^2(\boldsymbol{B}_{k+1}, \boldsymbol{B}^{\star}) &\leq \|\boldsymbol{B}_{k+1} - \mathcal{P}_{\boldsymbol{B}^{\star}}(\boldsymbol{B}_k)\|_F^2 \\
&\leq \|\widehat{\boldsymbol{B}}_{k+1} - \mathcal{P}_{\boldsymbol{B}^{\star}}(\boldsymbol{B}_k)\|_F^2 \\
&= \|\boldsymbol{B}_k - \mu\mathcal{G}(\boldsymbol{B}_k) - \mathcal{P}_{\boldsymbol{B}^{\star}}(\boldsymbol{B}_k)\|_F^2 \\
&= \|\boldsymbol{B}_k - \mathcal{P}_{\boldsymbol{B}^{\star}}(\boldsymbol{B}_k)\|_F^2 - 2\mu\langle \boldsymbol{B}_k - \mathcal{P}_{\boldsymbol{B}^{\star}}(\boldsymbol{B}_k), \mathcal{G}(\boldsymbol{B}_k)\rangle + \mu^2\|\mathcal{G}(\boldsymbol{B}_k)\|_F^2 \\
&\leq \operatorname{dist}^2(\boldsymbol{B}_k, \boldsymbol{B}^{\star}) - 2\alpha\mu\operatorname{dist}(\boldsymbol{B}_k, \boldsymbol{B}^{\star}) + \mu^2\xi^2,
\end{aligned}
\tag{12}
$$

where the second line utilizes (11), and the last line utilizes the Riemannian regularity condition (6).

It is clear from (12) that $\operatorname{dist}^2(\boldsymbol{B}_{k+1}, \boldsymbol{B}^{\star}) \leq \operatorname{dist}^2(\boldsymbol{B}_k, \boldsymbol{B}^{\star})$ if $\operatorname{dist}(\boldsymbol{B}_k, \boldsymbol{B}^{\star}) \geq \frac{\mu\xi^2}{2\alpha}$. In particular, when $\operatorname{dist}(\boldsymbol{B}_k, \boldsymbol{B}^{\star}) \geq \frac{\mu\xi^2}{\alpha}$, we have

$$
\begin{aligned}
\operatorname{dist}^2(\boldsymbol{B}_{k+1}, \boldsymbol{B}^{\star}) &\leq \operatorname{dist}^2(\boldsymbol{B}_k, \boldsymbol{B}^{\star}) - \alpha\mu\operatorname{dist}(\boldsymbol{B}_k, \boldsymbol{B}^{\star}) + \mu^2\xi^2 - \alpha\mu\operatorname{dist}(\boldsymbol{B}_k, \boldsymbol{B}^{\star}) \\
&\leq \left(\operatorname{dist}(\boldsymbol{B}_k, \boldsymbol{B}^{\star}) - \frac{\mu\alpha}{2}\right)^2,
\end{aligned}
$$

which implies that

$$
\operatorname{dist}(\boldsymbol{B}_{k+1}, \boldsymbol{B}^{\star}) \leq \operatorname{dist}(\boldsymbol{B}_k, \boldsymbol{B}^{\star}) - \frac{\mu\alpha}{2}
$$

since $\operatorname{dist}(\boldsymbol{B}_k, \boldsymbol{B}^{\star}) \geq \frac{\mu\xi^2}{\alpha} \geq \mu\alpha$.

On the other hand, when $\operatorname{dist}(\boldsymbol{B}_k, \boldsymbol{B}^{\star}) \leq \frac{\mu\xi^2}{\alpha}$, it also follows from (12) that

$$
\begin{aligned}
\operatorname{dist}^2(\boldsymbol{B}_{k+1}, \boldsymbol{B}^{\star}) &\leq \max\left\{\left(\frac{\mu\xi^2}{\alpha}\right)^2 - 2\mu\alpha\frac{\mu\xi^2}{\alpha} + \mu^2\xi^2, \mu^2\xi^2\right\} \\
&= \max\left\{\left(\frac{\mu\xi^2}{\alpha}\right)^2 - \mu^2\xi^2, \mu^2\xi^2\right\} \\
&\leq \max\left\{\left(\frac{\mu\xi^2}{\alpha}\right)^2 - \mu^2\xi^2, \mu^2\xi^2\frac{\xi^2}{\alpha^2}\right\} \\
&= \left(\frac{\mu\xi^2}{\alpha}\right)^2,
\end{aligned}
$$

where the first inequality follows that $h(t) := t^2 - 2\alpha\mu t$ is increasing in $[t', \infty]$ for any $t'$ such that $h(t') \geq 0$, and the second inequality utilizes (9). Thus by induction, (10) holds for all $k \geq 0$.

<div style="text-align: right">□</div>

# 3   Proof of Theorem 1

We first repeat Theorem 1 (which is a lightly stronger result than the one presented in the paper).

**Theorem 1.** *Suppose that $f$ satisfies the $(\alpha, \epsilon, \boldsymbol{B}^\star)$-RRC in Definition 1. Let $\{\boldsymbol{B}_k\}$ be the sequence generated by Algorithm 1 with step size*

$$\mu_k = \mu_0 \beta^k \tag{13}$$

*and initialization $\boldsymbol{B}_0$ satisfying $\operatorname{dist}(\boldsymbol{B}_0, \boldsymbol{B}^\star) \leq \epsilon$. Assume*

$$\mu_0 \leq \frac{\alpha \operatorname{dist}_0}{2\xi^2} \quad and \quad \sqrt{1 - 2\frac{\alpha\mu_0}{\operatorname{dist}_0} + \frac{\mu_0^2 \xi^2}{\operatorname{dist}_0^2}} =: \underline{\beta} \leq \beta < 1, \tag{14}$$

*for some $\operatorname{dist}(\boldsymbol{B}_0, \boldsymbol{B}^\star) \leq \operatorname{dist}_0 \leq \epsilon$, where $\xi$ is defined in (7). Then, the sequence $\{\boldsymbol{B}_k\}$ satisfies*

$$\operatorname{dist}(\boldsymbol{B}_k, \boldsymbol{B}^\star) \leq \operatorname{dist}_0 \beta^k \quad for\ all\ \ k \geq 0. \tag{15}$$

*Proof of Theorem 1.* We first show that $\underline{\beta}$ in (14) is well-defined and satisfies $0 < \underline{\beta} < 1$. To see this, on one hand, $\mu_0 \leq \alpha \operatorname{dist}_0/2\xi^2$ and (9) together imply $1 - 2\alpha\mu_0/\operatorname{dist}_0 \geq 0$. On the other hand, $-2\alpha\mu_0/\operatorname{dist}_0 + \mu_0^2\xi^2/\operatorname{dist}_0^2 < 0$ is a decreasing function of $\mu_0$ when $\mu_0 \in (0, \alpha \operatorname{dist}_0/2\xi^2]$. In particular, when $\mu_0 = \alpha \operatorname{dist}_0/2\xi^2$, we have $\underline{\beta} = \sqrt{1 - 3\alpha^2/4\xi^2}$, giving the fastest decaying rate by setting $\beta = \underline{\beta}$.

We now prove (15) by induction. It is clear that (15) holds when $k = 0$. Now assume that (15) holds at the $k$-th iteration, which implies that $\operatorname{dist}(\boldsymbol{B}_k, \boldsymbol{B}^\star) \leq \operatorname{dist}_0 \beta^k$. Since $\boldsymbol{B}_k$ satisfies the Riemannian regularity condition (6), we know that (12) holds:

$$\begin{aligned}
\operatorname{dist}^2(\boldsymbol{B}_{k+1}, \boldsymbol{B}^\star) &\leq \operatorname{dist}^2(\boldsymbol{B}_k, \boldsymbol{B}^\star) - 2\alpha\mu_k \operatorname{dist}(\boldsymbol{B}_k, \boldsymbol{B}^\star) + \mu_k^2 \xi^2 \\
&= (\operatorname{dist}(\boldsymbol{B}_k, \boldsymbol{B}^\star) - \alpha\mu_k)^2 + \mu_k^2(\xi^2 - \alpha^2).
\end{aligned} \tag{16}$$

From $\operatorname{dist}(\boldsymbol{B}_k, \boldsymbol{B}^\star) \leq \operatorname{dist}_0 \beta^k$ and

$$\operatorname{dist}_0 \beta^k \geq 2\frac{\mu_0 \xi^2}{\alpha} \beta^k \geq 2\alpha\mu_0 \beta^k = 2\alpha\mu_k \geq \alpha\mu_k,$$

where the first inequality is due to the assumption (14) and the second inequality follows $\xi \geq \alpha$ in (9). Therefore, (16) achieves its maximum at $\operatorname{dist}(\boldsymbol{B}_k, \boldsymbol{B}^\star) = \operatorname{dist}_0 \beta^k$. Plugging this observation into (16) gives

$$\begin{aligned}
\operatorname{dist}^2(\boldsymbol{B}_{k+1}, \boldsymbol{B}^\star) &\leq \operatorname{dist}_0^2 \beta^{2k} - 2\alpha\mu_k \operatorname{dist}_0 \beta^k + \mu_k^2 \xi^2 \\
&= \operatorname{dist}_0^2 \beta^{2k} - 2\alpha\mu_0 \operatorname{dist}_0 \beta^{2k} + \mu_0^2 \beta^{2k} \xi^2 \\
&= \operatorname{dist}_0^2 \beta^{2k} \left(1 - 2\frac{\alpha\mu_0}{\operatorname{dist}_0} + \frac{\mu_0^2 \xi^2}{\operatorname{dist}_0^2}\right) \\
&\leq \operatorname{dist}_0^2 \beta^{2(k+1)},
\end{aligned} \tag{17}$$

<div style="text-align: center">5</div>

where the last line holds because $\beta \geq \underline{\beta} = \sqrt{1 - 2\frac{\alpha\mu_0}{\mathrm{dist}_0} + \frac{\mu_0^2\xi^2}{\mathrm{dist}_0^2}}$. Hence, the proof is completed by induction. □

# 4 Proof of Theorem 2

We first repeat the DPCP problem. Given a dataset $\widetilde{\boldsymbol{\mathcal{X}}} = [\boldsymbol{\mathcal{X}} \; \boldsymbol{\mathcal{O}}]\boldsymbol{\Gamma} \in \mathbb{R}^{D \times L}$, where $\boldsymbol{\mathcal{X}} \in \mathbb{R}^{D \times N}$ are inlier points spanning a $d$-dimensional subspace $\mathcal{S}$ of $\mathbb{R}^D$, $\boldsymbol{\mathcal{O}}$ are outlier points, and $\boldsymbol{\Gamma}$ is an unknown permutation, the DPCP problem is

$$\underset{\boldsymbol{B} \in \mathbb{O}(c,D)}{\text{minimize}} f(\boldsymbol{B}) := \|\widetilde{\boldsymbol{\mathcal{X}}}^\top \boldsymbol{B}\|_{1,2} = \sum_{i=1}^{L} \|\widetilde{\boldsymbol{x}}_i^\top \boldsymbol{B}\|_2, \tag{18}$$

One choice for its Riemannian subgradient is

$$\mathcal{G}(\boldsymbol{B}) = (\mathbf{I} - \boldsymbol{B}\boldsymbol{B}^\top)\sum_{i=1}^{L} \widetilde{\boldsymbol{x}}_i \,\mathrm{sign}(\widetilde{\boldsymbol{x}}_i^\top \boldsymbol{B}), \quad \mathrm{sign}(\boldsymbol{a}) := \begin{cases} \boldsymbol{a}/\|\boldsymbol{a}\|_2 & \text{if } \boldsymbol{a} \neq \mathbf{0}, \\ \mathbf{0} & \text{if } \boldsymbol{a} = \mathbf{0}. \end{cases} \tag{19}$$

Let us recall several quantities: the first one, related to the outliers, characterizes the maximum Riemannian subgradient of $\frac{1}{M}\sum_{i=1}^{M}\|\boldsymbol{o}_i^\top \boldsymbol{B}\|_2$:

$$\eta_{\boldsymbol{\mathcal{O}}} := \frac{1}{M} \max_{\boldsymbol{B} \in \mathbb{O}(c,D)} \left\| (\mathbf{I} - \boldsymbol{B}\boldsymbol{B}^\top)\sum_{i=1}^{M} \boldsymbol{o}_i \,\mathrm{sign}(\boldsymbol{o}_i^\top \boldsymbol{B}) \right\|_F, \tag{20}$$

which appears in [6] when $\boldsymbol{B}$ is on $\mathbb{S}^{D-1}$. The second one is related to the inliers and is given by

$$c_{\boldsymbol{\mathcal{X}},\min} := \frac{1}{N} \min_{\boldsymbol{b} \in \mathcal{S} \cap \mathbb{S}^{D-1}} \|\boldsymbol{\mathcal{X}}^\top \boldsymbol{b}\|_1, \tag{21}$$

which is referred to as the *permeance statistic* in [7]. These quantities reflect how well distributed the inliers and outliers are, with larger values of $c_{\boldsymbol{\mathcal{X}},\min}$ (respectively, smaller values of $\eta_{\boldsymbol{\mathcal{O}}}$) corresponding to more uniform distributions of inliers (respectively, outliers).

We require one more result concerning the *principal angles* between two subspaces as follows.

**Definition 2.** *[8] Suppose $\boldsymbol{U} \in \mathbb{R}^{D \times p}$ and $\boldsymbol{V} \in \mathbb{R}^{D \times q}$ are two orthonormal bases. Suppose $p \geq q$. Then the principal angles between $\mathrm{Span}(\boldsymbol{U})$ and $\mathrm{Span}(\boldsymbol{V})$, $\phi_1(\boldsymbol{U},\boldsymbol{V}) \leq \phi_2(\boldsymbol{U},\boldsymbol{V}) \leq \cdots \leq \phi_q(\boldsymbol{U},\boldsymbol{V})$, are defined as*

$$\phi_i(\boldsymbol{U},\boldsymbol{V}) = \arccos(\sigma_i(\boldsymbol{U}^\top \boldsymbol{V}))$$

*for all $i \in \{1,2,\ldots,q\}$, where $\sigma_i(\cdot)$ denotes the $i$-th largest singular value. The largest principal angle $\phi_q(\boldsymbol{U},\boldsymbol{V})$ is referred to as the subspace angle between $\mathrm{Span}(\boldsymbol{U})$ and $\mathrm{Span}(\boldsymbol{V})$.*

**Lemma 1.** *[8] Suppose $\boldsymbol{U} \in \mathbb{R}^{D \times p}$ and $\boldsymbol{V} \in \mathbb{R}^{D \times q}$ are two orthonormal bases and $\begin{bmatrix} \boldsymbol{U} & \boldsymbol{U}^\perp \end{bmatrix}$ is an orthonormal basis of $\mathbb{R}^D$. Suppose $p \geq q$. Then the principal angles between $\mathrm{Span}(\boldsymbol{U})$ and $\mathrm{Span}(\boldsymbol{V})$ and the principal angles between $\mathrm{Span}(\boldsymbol{U}^\perp)$ and $\mathrm{Span}(\boldsymbol{V})$ satisfy the following relationship*

$$\left[ \frac{\pi}{2}, \ldots, \frac{\pi}{2}, \phi_q(\boldsymbol{U},\boldsymbol{V}), \ldots, \phi_1(\boldsymbol{U},\boldsymbol{V}) \right] = \left[ \frac{\pi}{2} - \phi_1(\boldsymbol{U}^\perp,\boldsymbol{V}), \ldots, \frac{\pi}{2} - \phi_q(\boldsymbol{U}^\perp,\boldsymbol{V}), 0, \ldots, 0 \right],$$

*where extra $\frac{\pi}{2}$'s and $0$'s are added on either side to match the sizes.*

We now review Theorem 2 and then prove it.

**Theorem 2** ($(\alpha, \epsilon, \boldsymbol{B}^\star)$-Riemannian regularity condition for the DPCP problem (18))**.** *For any* $\epsilon < \sqrt{2\left(1 - M\eta_{\mathcal{O}}/Nc_{\boldsymbol{\mathcal{X}},\min}\right)}$, *the DPCP problem* (18) *satisfies* $(\alpha, \epsilon, \boldsymbol{B}^\star)$-*Riemannian regularity condition with* $\boldsymbol{B}^\star = [\boldsymbol{S}^\perp]$ *and* $\alpha = ((1 - \epsilon^2/2)Nc_{\boldsymbol{\mathcal{X}},\min} - M\eta_{\mathcal{O}})/\sqrt{2c}$, *where* $\boldsymbol{S}^\perp$ *is an orthonormal basis for* $\mathcal{S}^\perp$. *Also,*

$$\|\mathcal{G}(\boldsymbol{B})\|_F \leq \sqrt{N}\,\|\boldsymbol{\mathcal{X}}\|_2 + M\eta_{\mathcal{O}}, \ \forall \boldsymbol{B} \in \mathbb{O}(c, D). \tag{22}$$

*Proof of Theorem 2.* We first establish the following result which is key to the Riemannian regularity condition for the DPCP problem.

**Lemma 2.** *Let* $\boldsymbol{B}^\star = [\boldsymbol{S}^\perp]$. *Then, for any* $\boldsymbol{B} \in \mathbb{O}(c, D)$, *we have*

$$\langle -\mathcal{G}(\boldsymbol{B}), \mathcal{P}_{\boldsymbol{B}^\star}(\boldsymbol{B}) - \boldsymbol{B} \rangle \geq \sin(\phi_{\max})(\cos(\phi_{\max})Nc_{\boldsymbol{\mathcal{X}},\min} - M\eta_{\mathcal{O}}), \tag{23}$$

*where* $\phi_{\max} := \phi_{\max}(\boldsymbol{B}, \mathcal{S}^\perp)$ *is the largest principal angle between* $\boldsymbol{B}$ *and* $\mathcal{S}^\perp$.

*Proof.* Let $\boldsymbol{S} \in \mathbb{R}^{D \times d}$ be an orthonormal basis of the subspace $\mathcal{S}$ and let $\boldsymbol{S}^\perp \in \mathbb{R}^{D \times c}$ be an orthonormal basis of the orthogonal complement $\mathcal{S}^\perp$. We rewrite $\boldsymbol{B}$ as

$$\boldsymbol{B} = \boldsymbol{S}\boldsymbol{S}^\top \boldsymbol{B} + \boldsymbol{S}^\perp (\boldsymbol{S}^\perp)^\top \boldsymbol{B},$$

where $\boldsymbol{S}\boldsymbol{S}^\top \boldsymbol{B}$ represents the projection of $\boldsymbol{B}$ onto the subspace $\mathcal{S}$, while the other term $\boldsymbol{S}^\perp (\boldsymbol{S}^\perp)^\top \boldsymbol{B}$ represents the projection of $\boldsymbol{B}$ onto the complement $\mathcal{S}^\perp$. Let $(\boldsymbol{S}^\perp)^\top \boldsymbol{B} = \boldsymbol{U}\cos(\boldsymbol{\Phi})\boldsymbol{R}^\top$ be the canonical SVD of $(\boldsymbol{S}^\perp)^\top \boldsymbol{B}$, where $\cos(\boldsymbol{\Phi})$ is a diagonal matrix with $\cos(\phi_1), \ldots, \cos(\phi_c)$ along its diagonal, $\boldsymbol{U} \in \mathbb{R}^{c \times c}, \boldsymbol{R} \in \mathbb{R}^{c \times c}$ are orthonormal matrices. Here $\phi_i$ is the $i$-th principal angles between $\boldsymbol{B}$ and $\mathcal{S}^\perp$. When $\phi_1 = \cdots = \phi_c = 0$, it implies that $\boldsymbol{B} \in [\boldsymbol{S}^\perp]$, i.e., $\boldsymbol{B}$ is equivalent to $\boldsymbol{S}^\perp$.

Without loss of generality, we assume $c \leq d$.[1] In this case, we can then rewrite $\boldsymbol{S}^\top \boldsymbol{B} = \boldsymbol{V}\sin(\boldsymbol{\Phi})\boldsymbol{R}^\top$, where $\boldsymbol{V} \in \mathbb{R}^{d \times c}$ is an orthonormal matrix. Thus, we have

$$\boldsymbol{B} = \boldsymbol{S}\boldsymbol{V}\sin(\boldsymbol{\Phi})\boldsymbol{R}^\top + \boldsymbol{S}^\perp \boldsymbol{U}\cos(\boldsymbol{\Phi})\boldsymbol{R}^\top. \tag{24}$$

After defining

$$\boldsymbol{G} = \boldsymbol{S}\boldsymbol{V}\cos(\boldsymbol{\Phi})\sin(\boldsymbol{\Phi})\boldsymbol{R}^\top - \boldsymbol{S}^\perp \boldsymbol{U}\sin^2(\boldsymbol{\Phi})\boldsymbol{R}^\top, \tag{25}$$

we have

$$
\begin{aligned}
\langle -\mathcal{G}(\boldsymbol{B}), \mathcal{P}_{\boldsymbol{B}^\star}(\boldsymbol{B}) - \boldsymbol{B} \rangle &= \langle -\mathcal{G}(\boldsymbol{B}), \boldsymbol{S}^\perp \boldsymbol{U}\boldsymbol{R}^\top \rangle \\
&= -\left\langle (\mathbf{I} - \boldsymbol{B}\boldsymbol{B}^\top)\left(\sum_{j=1}^{L} \widetilde{\boldsymbol{x}}_j \operatorname{sign}(\widetilde{\boldsymbol{x}}_j^\top \boldsymbol{B})\right), \boldsymbol{S}^\perp \boldsymbol{U}\boldsymbol{R}^\top \right\rangle \\
&= \left\langle \sum_{j=1}^{N} \boldsymbol{x}_j \operatorname{sign}(\boldsymbol{x}_j^\top \boldsymbol{B}) + \sum_{j=1}^{M} \boldsymbol{o}_j \operatorname{sign}(\boldsymbol{o}_j^\top \boldsymbol{B}), \boldsymbol{G} \right\rangle \\
&= \sum_{j=1}^{N} \left\langle \boldsymbol{x}_j^\top \boldsymbol{S}\boldsymbol{V}\cos(\boldsymbol{\Phi})\sin(\boldsymbol{\Phi}), \operatorname{sign}(\boldsymbol{x}_j^\top \boldsymbol{S}\boldsymbol{V}\sin(\boldsymbol{\Phi})) \right\rangle - \left\langle \boldsymbol{G}, (\mathbf{I} - \boldsymbol{B}\boldsymbol{B}^\top)\sum_{j=1}^{M} \boldsymbol{o}_j \operatorname{sign}(\boldsymbol{o}_j^\top \boldsymbol{B}) \right\rangle
\end{aligned}
\tag{26}
$$

where the very last line utilizes $\text{sign}(\boldsymbol{a}\boldsymbol{R}^\top) = \text{sign}(\boldsymbol{a})\boldsymbol{R}^\top$ ($\boldsymbol{R} \in \mathbb{R}^{c \times c}$ is an orthonormal matrix) and the fact that $\boldsymbol{G} \in \text{Span}(\boldsymbol{B}^\perp)$.

We now bound the first term in the last line of (26) by

$$\sum_{j=1}^{N} \left\langle \boldsymbol{x}_j^\top \boldsymbol{S}\boldsymbol{V} \sin(\boldsymbol{\Phi}) \cos(\boldsymbol{\Phi}), \text{sign}(\boldsymbol{x}_j^\top \boldsymbol{S}\boldsymbol{V} \sin(\boldsymbol{\Phi})) \right\rangle$$

$$\geq \cos(\phi_c) \sum_{j=1}^{N} \left\langle \boldsymbol{x}_j^\top \boldsymbol{S}\boldsymbol{V} \sin(\boldsymbol{\Phi}), \text{sign}(\boldsymbol{x}_j^\top \boldsymbol{S}\boldsymbol{V} \sin(\boldsymbol{\Phi})) \right\rangle$$

$$= \cos(\phi_c) \sum_{j=1}^{N} \left\| \boldsymbol{x}_j^\top \boldsymbol{S}\boldsymbol{V} \sin(\boldsymbol{\Phi}) \right\|_2 \geq \cos(\phi_c) \sin(\phi_c) \sum_{j=1}^{N} |\boldsymbol{x}_j^\top \boldsymbol{S}\boldsymbol{v}_c|$$

$$\geq \cos(\phi_c) \sin(\phi_c) N c_{\boldsymbol{\mathcal{X}},\min},$$

where the first inequality follows because $0 \leq \phi_1 \leq \phi_2 \leq \cdots \phi_c \leq \frac{\pi}{2}$, and the last inequality utilizes (21) since $\boldsymbol{S}\boldsymbol{v}_c \in \mathcal{S} \cap \mathbb{S}^{D-1}$. On the other hand, the second term in the last line of (26) can be bounded by

$$\left| \left\langle \boldsymbol{G}, (\mathbf{I} - \boldsymbol{B}\boldsymbol{B}^\top) \sum_{j=1}^{M} \boldsymbol{o}_j \text{sign}(\boldsymbol{o}_j^\top \boldsymbol{B}) \right\rangle \right|$$

$$= \left| \left\langle \boldsymbol{S}\boldsymbol{V} \cos(\boldsymbol{\Phi}) \sin(\boldsymbol{\Phi})\boldsymbol{R}^\top - \boldsymbol{S}^\perp \boldsymbol{U} \sin^2(\boldsymbol{\Phi})\boldsymbol{R}^\top, (\mathbf{I} - \boldsymbol{B}\boldsymbol{B}^\top) \sum_{j=1}^{M} \boldsymbol{o}_j \text{sign}(\boldsymbol{o}_j^\top \boldsymbol{B}) \right\rangle \right|$$

$$\leq \sin(\phi_c) \left| \left\langle \boldsymbol{S}\boldsymbol{V} \cos(\boldsymbol{\Phi})\boldsymbol{R}^\top - \boldsymbol{S}^\perp \boldsymbol{U} \sin(\boldsymbol{\Phi})\boldsymbol{R}^\top, (\mathbf{I} - \boldsymbol{B}\boldsymbol{B}^\top) \sum_{j=1}^{M} \boldsymbol{o}_j \text{sign}(\boldsymbol{o}_j^\top \boldsymbol{B}) \right\rangle \right|$$

$$\leq \sin(\phi_c) \left\| (\mathbf{I} - \boldsymbol{B}\boldsymbol{B}^\top) \sum_{j=1}^{M} \boldsymbol{o}_j \text{sign}(\boldsymbol{o}_j^\top \boldsymbol{B}) \right\|_F \leq \sin(\phi_c) M \eta_{\boldsymbol{\mathcal{O}}}$$

where the first inequality follows because $0 \leq \phi_1 \leq \phi_2 \leq \cdots \phi_c \leq \frac{\pi}{2}$, the second inequality utilizes the fact that $\boldsymbol{S}\boldsymbol{V} \cos(\boldsymbol{\Phi})\boldsymbol{R}^\top - \boldsymbol{S}^\perp \boldsymbol{U} \sin(\boldsymbol{\Phi})\boldsymbol{R}^\top$ is an orthonormal matrix, and the last inequality follows from (20). This completes the proof. $\qquad\square$

We now turn to prove the $(\alpha, \epsilon, \boldsymbol{B}^\star)$-Riemannian regularity condition. First note that

$$\|\mathcal{P}_{\boldsymbol{B}^\star}(\boldsymbol{B}) - \boldsymbol{B}\|_F^2 = 2 \sum_{i=1}^{c} (1 - \cos(\phi_i)) \leq 2c(1 - \cos(\phi_c))$$

$$= 4c \sin^2\left(\frac{\phi_c}{2}\right) \leq 2c \sin^2(\phi_c), \tag{27}$$

where the last inequality utilizes $\sin(\alpha) \geq \sqrt{2} \sin(\alpha/2)$ for any $\alpha \in [0, \frac{\pi}{2}]$. Combining (27) together with (23) give

$$\langle -\mathcal{G}(\boldsymbol{B}), \mathcal{P}_{\boldsymbol{B}^\star}(\boldsymbol{B}) - \boldsymbol{B} \rangle \geq \frac{\cos(\phi_c) N c_{\boldsymbol{\mathcal{X}},\min} - M \eta_{\boldsymbol{\mathcal{O}}}}{\sqrt{2c}} \|\mathcal{P}_{\boldsymbol{B}^\star}(\boldsymbol{B}) - \boldsymbol{B}\|_F. \tag{28}$$

On the other hand, we have

$$\|\boldsymbol{B} - \mathcal{P}_{\boldsymbol{B}^\star}(\boldsymbol{B})\|_F^2 = 2\sum_{i=1}^c (1 - \cos(\phi_i)) \geq 2(1 - \cos(\phi_c)),$$

which implies that $\cos(\phi_c) \geq 1 - \frac{\|\boldsymbol{B} - \mathcal{P}_{\boldsymbol{B}^\star}(\boldsymbol{B})\|_F^2}{2}$. This together with (28) and $\|\boldsymbol{B} - \mathcal{P}_{\boldsymbol{B}^\star}(\boldsymbol{B})\|_F \leq \epsilon$ complete the proof of the $(\alpha, \epsilon, \boldsymbol{B}^\star)$-Riemannian regularity condition. The rest is to prove (22).

**Proof of** (22)    For convenience, denote by $\operatorname{sign}(\boldsymbol{\mathcal{X}}^\top \boldsymbol{B}) = \begin{bmatrix} \operatorname{sign}(\boldsymbol{x}_1^\top \boldsymbol{B}) & \cdots & \operatorname{sign}(\boldsymbol{x}_N^\top \boldsymbol{B}) \end{bmatrix}^\top$. Using (20) and the fact that $\|\operatorname{sign}(\boldsymbol{\mathcal{X}}^\top \boldsymbol{B})\|_F \leq \sqrt{N}$ allows us to bound the Riemannian subgradient in (19) as

$$
\begin{aligned}
\|\mathcal{G}(\boldsymbol{B})\|_F &\leq \left\|(\mathbf{I} - \boldsymbol{B}\boldsymbol{B}^\top)\boldsymbol{\mathcal{X}} \operatorname{sign}(\boldsymbol{\mathcal{X}}^\top \boldsymbol{B})\right\|_F + \left\|(\mathbf{I} - \boldsymbol{B}\boldsymbol{B}^\top)\sum_{j=1}^M \boldsymbol{o}_j \operatorname{sign}(\boldsymbol{o}_j^\top \boldsymbol{B})\right\|_F \\
&\leq \left\|(\mathbf{I} - \boldsymbol{B}\boldsymbol{B}^\top)\boldsymbol{\mathcal{X}}\right\|_2 \left\|\operatorname{sign}(\boldsymbol{\mathcal{X}}^\top \boldsymbol{B})\right\|_F + M\eta_{\boldsymbol{\mathcal{O}}} \\
&\leq \sqrt{N}\|\boldsymbol{\mathcal{X}}\|_2 + M\eta_{\boldsymbol{\mathcal{O}}},
\end{aligned}
$$

where the second inequality follows from $\|\boldsymbol{A}\boldsymbol{B}\|_F \leq \|\boldsymbol{A}\|_2\|\boldsymbol{B}\|_F$.

□

# 5    Proof of Corollary 2

**Definition 3** (Random model for ODL [9])**.** *Assume $\boldsymbol{A} \in \mathbb{R}^{D \times D}$ is a fixed but unknown orthonormal matrix. The data is generated as $\widetilde{\boldsymbol{\mathcal{X}}} = \boldsymbol{A}\boldsymbol{S}$, where each column of $\boldsymbol{S} \in \mathbb{R}^{D \times D}$ is an i.i.d. Bernoulli-Gaussian random vector with parameter $\rho \in (0, 1)$ that controls the sparsity.*

We first repeat the Riemannian regularity condition for the ODL problem.

**Theorem 3.** *[9, Theorem 3.6] Assume $\rho \in [1/D, 1/2]$ in the random model of Definition 3. There exist universal constants $C, c > 0$ such that if $N \geq CD^4\zeta^{-2}\rho^{-2}\log(D/\zeta)$, $\forall \zeta \in (0, 1)$, then with probability at least $1 - \exp(-cN\rho^3\zeta^2 D^{-3}/\log N)$ the ODL problem satisfies (6) for any $\boldsymbol{b} \in \mathcal{I}_\zeta^i$ with $\mathcal{G}(\boldsymbol{b})$ and $\boldsymbol{B}^\star = \boldsymbol{e}_i$ for any $i$, and*

$$\alpha = \tfrac{1}{16}\rho(1-\rho)\zeta D^{-\frac{3}{2}}. \tag{29}$$

Now we repeat Corollary 2 and then prove it.

**Corollary 2.** *Let $\{\boldsymbol{b}_k\}$ be the sequence generated by Algorithm 1 for the ODL problem with $\boldsymbol{b}_0 \in \mathcal{I}_\zeta^i$ ($\zeta \leq \frac{55}{64}$) and step size $\mu_k = \mu_0\beta^k$, where $\mu_0$ and $\beta$ satisfy the conditions in Theorem 1 with $\xi = 2$ and $\epsilon = \sqrt{2}$, and $\alpha = \frac{1}{16}\rho(1-\rho)\zeta D^{-\frac{3}{2}}$. Under the same setup as in Theorem 3, with probability at least $1 - \exp(-cN\rho^3\zeta^2 D^{-3}/\log N)$, $\{\boldsymbol{b}_k\}$ converges to $\boldsymbol{e}_i$ at an R-linear rate, i.e.,*

$$\operatorname{dist}(\boldsymbol{b}_k, \boldsymbol{e}_i) \leq \beta^k \operatorname{dist}(\boldsymbol{b}_0, \boldsymbol{e}_i). \tag{30}$$

*Proof.* To apply Theorem 1, we require an upper bound on the norm of the Riemannian subgradients. It follows from [9, Proposition 3.7] that if $N \geq CD \log D$, then $\sup_{\boldsymbol{B} \in \mathbb{S}^{D-1}} \|\mathcal{G}(\boldsymbol{B})\|_2 \leq 2$ with probability at least $1 - \exp(-cN\rho/\log N)$. We also require $\boldsymbol{b}_k \in \mathcal{I}_\zeta^i$ for all $k$ so that the Riemannian regularity condition (6) holds at all the iterates. A sufficient condition to guarantee $\boldsymbol{b}_k \in \mathcal{I}_\zeta^i$ is that $\mu_k \leq \min\{\frac{1}{100}, \frac{1-\zeta}{9}\} \frac{1}{D^{1/2}}$ [9, Proposition D.2]. Plugging (29), $\epsilon = \sqrt{2}$, and $\xi = 2$ into (14) gives $\mu_k \leq \mu_0 \leq \frac{\zeta}{64 D^{3/2}} \leq \min\{\frac{1}{100}, \frac{1-\zeta}{9}\} \frac{1}{D^{1/2}}$ since $\zeta \leq \frac{55}{64}$. $\qquad\square$

# 6    Initialization

**Lemma 3.** *Consider a spectral initialization $\boldsymbol{B}_0$ by taking the bottom $c$ eigenvectors of $\widetilde{\boldsymbol{\mathcal{X}}}\widetilde{\boldsymbol{\mathcal{X}}}^\top$. Then, it satisfies*

$$\|\boldsymbol{B}_0 - \mathcal{P}_{\boldsymbol{B}^\star}(\boldsymbol{B}_0)\|_F^2 \leq \frac{\sum_{j=1}^c \sigma_j^2(\boldsymbol{\mathcal{O}}) - \sum_{j=D-c+1}^D \sigma_j^2(\boldsymbol{\mathcal{O}})}{\sigma_d^2(\boldsymbol{\mathcal{X}})}, \tag{31}$$

*where $\sigma_\ell$ denotes the $\ell$-th largest singular value.*

*Proof.* Note that for any $\boldsymbol{B} \perp \mathcal{S}$, $\|\widetilde{\boldsymbol{\mathcal{X}}}^\top \boldsymbol{B}\|_F^2 = \|\boldsymbol{\mathcal{O}}^\top \boldsymbol{B}\|_F^2 = \operatorname{trace}(\boldsymbol{B}^\top \boldsymbol{\mathcal{O}} \boldsymbol{\mathcal{O}}^\top \boldsymbol{B}) = \sum_{j=1}^c \boldsymbol{b}_j^\top \boldsymbol{\mathcal{O}} \boldsymbol{\mathcal{O}}^\top \boldsymbol{b}_j \leq \sum_{j=1}^c \sigma_j^2(\boldsymbol{\mathcal{O}})$. Thus, since $\boldsymbol{B}_0$ is the optimal solution to $\arg \min_{\boldsymbol{B} \in \mathbb{O}(c,D)} \|\widetilde{\boldsymbol{\mathcal{X}}}^\top \boldsymbol{B}\|_F^2$, we have

$$\|\widetilde{\boldsymbol{\mathcal{X}}}^\top \boldsymbol{B}_0\|_F^2 \leq \sum_{j=1}^c \sigma_j^2(\boldsymbol{\mathcal{O}}).$$

On the other hand, let $\boldsymbol{S}$ be an orthonormal basis for $\mathcal{S}$ and let $\boldsymbol{\Theta}$ be the coefficients of $\boldsymbol{\mathcal{X}}$ in $\boldsymbol{S}$, i.e., $\boldsymbol{\mathcal{X}} = \boldsymbol{S}\boldsymbol{\Theta}$, we have

$$\|\widetilde{\boldsymbol{\mathcal{X}}}^\top \boldsymbol{B}_0\|_F^2 = \|\boldsymbol{\mathcal{X}}^\top \boldsymbol{B}_0\|_F^2 + \|\boldsymbol{\mathcal{O}}^\top \boldsymbol{B}_0\|_F^2 = \|\boldsymbol{\mathcal{X}}^\top \boldsymbol{S}\boldsymbol{S}^\top \boldsymbol{B}_0\|_F^2 + \|\boldsymbol{\mathcal{O}}^\top \boldsymbol{B}_0\|_F^2$$

$$= \|\boldsymbol{\Theta}^\top \boldsymbol{S}^\top \boldsymbol{B}_0\|_F^2 + \|\boldsymbol{\mathcal{O}}^\top \boldsymbol{B}_0\|_F^2 \geq \sigma_{\min}^2(\boldsymbol{\Theta})\|\boldsymbol{S}^\top \boldsymbol{B}_0\|_F^2 + \|\boldsymbol{\mathcal{O}}^\top \boldsymbol{B}_0\|_F^2$$

$$\geq \sigma_d^2(\boldsymbol{\mathcal{X}})\|\boldsymbol{B}_0 - \mathcal{P}_{\boldsymbol{B}^\star}(\boldsymbol{B}_0)\|_F^2 + \sum_{j=D-c+1}^D \sigma_j^2(\boldsymbol{\mathcal{O}}),$$

where we first utilize the fact that $\boldsymbol{\mathcal{X}}$ lies in $\mathcal{S}$ such that $\boldsymbol{\mathcal{X}} = \boldsymbol{S}\boldsymbol{S}^\top \boldsymbol{\mathcal{X}}$, the inequality follows because $\|\boldsymbol{A}\boldsymbol{B}\|_F^2 = \operatorname{trace}(\boldsymbol{A}^\top \boldsymbol{A} \boldsymbol{B}\boldsymbol{B}^\top) \geq \sigma_{\min}(\boldsymbol{A}^\top \boldsymbol{A})\|\boldsymbol{B}\boldsymbol{B}^\top\|_F$ for any $\boldsymbol{A}, \boldsymbol{B}$, and the last line follows from $\|\boldsymbol{S}^\top \boldsymbol{B}_0\|_F^2 = \|\boldsymbol{S}\boldsymbol{S}^\top \boldsymbol{B}_0\|_F^2 = \|\boldsymbol{B}_0 - \boldsymbol{S}^\perp (\boldsymbol{S}^\perp)^\top \boldsymbol{B}_0\|_F^2 = \|\boldsymbol{B}_0 - \mathcal{P}_{\boldsymbol{B}^\star}(\boldsymbol{B}_0)\|_F^2$. Combining the above two equations gives

$$\|\boldsymbol{B}_0 - \mathcal{P}_{\boldsymbol{B}^\star}(\boldsymbol{B}_0)\|_F^2 \leq \frac{\sum_{j=1}^c \sigma_j^2(\boldsymbol{\mathcal{O}}) - \sum_{j=D-c+1}^D \sigma_j^2(\boldsymbol{\mathcal{O}})}{\sigma_d^2(\boldsymbol{\mathcal{X}})}.$$

$\qquad\square$

# 7    Random Spherical Model

In this section, we consider the following random spherical model.

**Definition 1.** *For any given subspace $\mathcal{S}$ of dimension $d < D$, a random spherical model refers to that the columns of $\mathcal{O}$ drawn independently and uniformly at random from the unit sphere $\mathbb{S}^{D-1}$, and the columns of $\mathcal{X}$ are drawn independently and uniformly at random from the intersection of the unit sphere with the subspace $\mathcal{S}$.*

We require the following result from [10, Lemma 4] concerning $c_{\mathcal{X},\min}$.

**Lemma 4.** *[10, Lemma 4] Under the random spherical model in Definition 1, we have*

$$\mathbb{P}\left(c_{\mathcal{X},\min} \geq c_d - (2 + \frac{t}{2})/\sqrt{N}\right) \geq 1 - 2e^{-\frac{t^2}{2}},$$

*where*

$$c_D := \frac{(D-2)!!}{(D-1)!!} \cdot \begin{cases} \frac{2}{\pi}, & D \text{ is even,} \\ 1, & D \text{ is old,} \end{cases} \quad \text{where } k!! := \begin{cases} k(k-2)(k-4)\cdots 4 \cdot 2, & k \text{ is even,} \\ k(k-2)(k-4)\cdots 3 \cdot 1, & k \text{ is old.} \end{cases} \quad (32)$$

**Lemma 5.** *Let $\boldsymbol{o}_1, \ldots, \boldsymbol{o}_M$ be uniformly distributed on $\mathbb{S}^{D-1}$. Then for any $t > 0$*

$$\mathbb{P}\left[\eta_{\mathcal{O}} \gtrsim \frac{\sqrt{cD}\log(c_D D) + t}{\sqrt{M}}\right] \leq 2\exp(-t^2/2), \quad (33)$$

*where $c = D - d$ is the co-dimensions.*

Its proof is given in Section 10.

The following results provide concentration inequalities for the singular values appeared in (31) when the inliners and outliers are generated from a random spherical model.

**Lemma 6.** *[11, Theorem 5.39] Under the random spherical model in Definition 1, then for every $t > 0$, there exist constants $C_1, C_2$ such that*

$$\mathbb{P}\left(\sigma_1(\mathcal{O}) \geq \frac{\sqrt{M} + C_2\sqrt{D} + t}{\sqrt{D}}\right) \leq e^{-C_1 t^2},$$

$$\mathbb{P}\left(\sigma_D(\mathcal{O}) \leq \frac{\sqrt{M} - C_2\sqrt{D} - t}{\sqrt{D}}\right) \leq e^{-C_1 t^2}, \quad (34)$$

$$\mathbb{P}\left(\sigma_d(\mathcal{X}) \leq \frac{\sqrt{N} - C_2\sqrt{d} - t}{\sqrt{d}}\right) \leq e^{-C_1 t^2}.$$

The following result establishes that the spectral initialization satisfies the condition $\mathrm{dist}^2(\boldsymbol{B}_0, \boldsymbol{S}^\perp) < 2\left(1 - M\eta_{\mathcal{O}}/Nc_{\mathcal{X},\min}\right)$ with high probability when the data are generated from a random spherical model.

**Corollary 3.** *Consider the same random spherical model as in Definition 1. Then for any positive number $t < \min\{\sqrt{N} - C_2\sqrt{d}, 2c_d\sqrt{N} - 4\}$, with probability at least $1 - 4e^{-t^2/2} - 3e^{-C_1 t^2}$, the spectral initialization $\boldsymbol{B}_0$ in Lemma 3 satisfies the condition $\mathrm{dist}^2(\boldsymbol{B}_0, \boldsymbol{S}^\perp) < 2\left(1 - M\eta_{\mathcal{O}}/Nc_{\mathcal{X},\min}\right)$,*

*provided that*

$$\frac{2cd\sqrt{M}(C_2\sqrt{D}+t)}{D(\sqrt{N}-C_2\sqrt{d}-t)^2} + \frac{C_0(\sqrt{cMD}\log(c_DD)+\sqrt{M}t)}{c_dN-(2+\frac{t}{2})\sqrt{N}} < 1,$$
$$c_dN-(2+t/2)\sqrt{N} > C_0\left(\sqrt{cD}\log(c_DD)+t\right)\sqrt{M},$$

$$(35)$$

*where $c_d$ and $c_D$ are defined in (32), and $C_0, C_1$ and $C_2$ are universal constants indepedent of $N, M, D, d$ and $t$.*

*Proof.* This follows by combining Lemma 3, Lemma 4, Lemma 5, and Lemma 6.  □

Note that the first line in (35) suggests $O\left(\frac{cd\sqrt{M}}{\sqrt{D}N} + \frac{\sqrt{cD}\log D\sqrt{M}}{N}\right) < 1$, while the seond line of (35) suggests that $O\left(\frac{\sqrt{cD}\log D\sqrt{M}}{N}\right) < 1$. The combination of both implies that the projected Riemannian subgradient method with a spectral initialization can converge to $\boldsymbol{S}^\perp$ in a linear rate when there are $M = O\left(\frac{D}{c^2(d+D\log D)^2}N^2\right)$ outliers.

# 8  Comparision with the Regularity Condition for Smooth Function

Aside from the weak convexity and shaprness, another regularity condition related to Definition 1 is the one proposed in [12]: we say a continuously differentiable function $g$ satisfies the $(\alpha, \gamma, \epsilon)$-regularity condition if for all $\boldsymbol{x} \in \mathbb{R}^D$ such that $\text{dist}(\boldsymbol{x}, \mathcal{X}) \leq \epsilon$, we have

$$\langle \mathcal{P}_\mathcal{X}(\boldsymbol{x}) - \boldsymbol{x}, -\nabla g(\boldsymbol{x}) \rangle$$
$$\geq \alpha \, \text{dist}^2(\boldsymbol{x}, \mathcal{X}) + \gamma \|\nabla g(\boldsymbol{x})\|^2.$$

$$(36)$$

We now compare (6) with (36). On one hand, (36) has similar form to (6) as both attempt to provide lower bounds for the inner product between the gradient (or Riemannian subgradient) and the vector $\boldsymbol{x} - \mathcal{P}_\mathcal{X}(\boldsymbol{x})$ for any $\boldsymbol{x}$ that is close to $\mathcal{X}$. On the other hand, (36) mainly differs from (6) in two aspects. First note that unlike (36), there is no $\|\mathcal{G}(\boldsymbol{B})\|$ term (which is $\|\nabla g(\boldsymbol{x})\|^2$ in (36)) in (6). This is mainly because as we illustrated before, the Riemannian subgradient does not vanish even when $\boldsymbol{B}$ approaching $\boldsymbol{B}^\star$. Thus, it is impossible to include the $\|\mathcal{G}(\boldsymbol{B})\|$ term into (6) as its left hand side (LHS) goes to 0 when $\boldsymbol{B}$ tends to $\boldsymbol{B}^\star$. Besides, (6) involves the term $\text{dist}(\boldsymbol{B}, \boldsymbol{B}^\star)$, while (36) has the term $\text{dist}^2(\boldsymbol{x}, \mathcal{X})$. If we apply the Cauchy-Schwarz inequality to the LHS of (36), we obtain

$$\gamma \|\nabla g(\boldsymbol{x})\| \leq \text{dist}(\boldsymbol{x}, \mathcal{X}) - \frac{\text{dist}^2(\boldsymbol{x}, \mathcal{X})}{\|\nabla g(\boldsymbol{x})\|},$$

which implies $\nabla g(\boldsymbol{x}) \to \boldsymbol{0}$ when $\text{dist}(\boldsymbol{x}, \mathcal{X}) \to 0$. This is in sharp contrast to (8). Informally speaking, the regularity condition in (36) describes certain geometric property of smooth functions while the Riemannian regularity condition in Definition 1 characterizes certain geometric property of non-smooth functions.

(a) $D = 30, N = 500, \gamma = 0.7$  (b) $D = 30, d = 25, N = 500$  (c) $D = 30, d = 25, M = 100$

Figure 1: The convergence of Algorithm 1 (the Riemannian SubGM with geometrically diminishing step size) for the DPCP problems with $\beta = 0.8$ and $\mu_0$ determined by line search method. Here $\gamma = \frac{M}{M+N}$ denotes the outlier ratio.

# 9    Additional Experiments

## 9.1    DPCP for Robust Subspace Learning

We use synthetic experiments under different settings to further verify the projected Riemannian subgradient method with geometrically diminishing step sizes for the DPCP problem. We randomly sample a subspace $\mathcal{S}$ of dimension $d < D - 1$, and uniformly at random sample $N$ inliers and $M$ outliers with unit $\ell_2$-norm. Denote by $\gamma = \frac{M}{M+N}$ the outlier ratio. We set $\beta = 0.8$ for geometrically diminishing step size with initial step size obtained by one iteration of a backtracking line search. We define $\boldsymbol{B}_0$ to be the bottom eigenvector of $\widetilde{\boldsymbol{\mathcal{X}}}\widetilde{\boldsymbol{\mathcal{X}}}^\top$. Figure 1 displays the convergence of $\theta$ (to 0) with different $d$, $N$ and outlier ratio $\gamma$. In particular, Figure 1a shows the convergence of $\theta$ with $D = 30, N = 500, \gamma = 0.7$ and different subspace dimension $d$. We obsrve linear convergence in this case, irrespectively the subspace dimension $d$. In Figure 1b, we set $D = 30, d = 25, N = 500$ and vary the outlier ratio $\gamma$ from 0.1 to 0.9. We also observe linear convergence expept for the case $\gamma = 0.9$, in which we have much more outliers than inliers. Finally we display experiments with varied $N$ in Figure 1c. We also observe linear convergence for Algorithm 1 given sufficient number of inliers.

## 9.2    Orthogonal Dictionary Learning

As illustrated in the paper, we first generate a random orthogonal dictionary $\boldsymbol{A} \in \mathbb{R}^{D \times D}$ with $D = 70$. Set the sparsity level $\rho = 0.3$ and the number of data points $N \approx 10D^{1.5} = 5857$. The initialization $\boldsymbol{B}_0$ is randomly generated from the unit sphere $\mathbb{S}^{D-1}$. Figure 2 displays the effect of the initial step size $\mu_0$ and the decaying factor $\beta$ for Algorithm 1 with geometrically diminishing step size $\mu_k = \mu_0 \beta^k$. We observe similar phenomena as for the DPCP problem. First observe from Figure 2a that, as expected, $\beta$ controls the convergence speed: a value of $\beta$ too small ($\beta = 0.7$) may result in no convergence, in agreement with (14) and (15); whereas when $\beta \geq 0.8$, the algorithm converges in a linear rate, with a larger value of $\beta$ resulting in a slower convergence speed (comparing $\beta = 0.8, 0.9, 0.95$). Figure 2b displays the effect of $\mu_0$ when $\beta$ is fixed. We observe that a value of $\mu_0$ too small ($\mu_0 = 1$) results in no convergence. This can be explained following the discussion after Theorem 1: when $\mu_0$ is small, then the smallest allowable decaying factor $\underline{\beta}$ in (14) increases

when $\mu_0$ decreases and particularly $\beta \to 1$ when $\mu_0 \to 0$, thus contradicting the requirement $\beta \geq \underline{\beta}$ in (14) when we fix $\beta = 0.9$ and set $\overline{\mu_0}$ too small ($\mu_0 = 1$).

(a) $\mu_0 = 10$            (b) $\beta = 0.9$

Figure 2: Convergence of Algorithm 1 with different initial step size $\mu_0$ and decaying factor $\beta$ for dictionary learning.

# 10 Proof of Lemma 5

## 10.1 Preliminaries

Suppose $X_1, \ldots, X_n$ are $n$ independent and identically distributed (i.i.d.) random observations from a probability measure $P$ on a measurable space $(\mathcal{X}, \mathcal{A})$. Given a measurable function $f : \mathcal{X} \to \mathbb{R}$, the *empirical process* evaluated at $f$ is defined as

$$\mathbb{G}_n f := \sqrt{n} \left( \frac{1}{n} \sum_{i=1}^{n} f(X_i) - \int f \, \mathrm{d} P \right), \tag{37}$$

where $\int f \, \mathrm{d} P$ is the expectation of $f$ under $P$ and $\frac{1}{n} \sum_{i=1}^{n} f(X_i)$ is called the *empirical distribution*. There are several results concerning the supreme of $\mathbb{G}_n f$ over a given class $\mathcal{F}$ of measurable functions.

Define an *envelope function* $F : \mathcal{X} \to \mathbb{R}$ such that $|f| \leq F$ for every $f \in \mathcal{F}$. The $L_r(P)$-norm is defined as $\|f\|_{L_r(P)} = (\int |f|^r \, \mathrm{d} P)^{1/r}$. We need one more definition for the so-called *bracket number* which (informally speaking) measures the size of a class functions $\mathcal{F}$. Given two functions $l$ and $u$, the *bracket* $[l, u]$ is the set of all functions $f$ with $l \leq f \leq u$. An $\epsilon$-bracket in $L_r(P)$ is a bracket $[l, u]$ with $\int (u - l)^r \, \mathrm{d} P \leq \epsilon^r$ (since $l \leq u$, it is equivalent to say $\|u - l\|_{L_r(P)} \leq \epsilon$). The bracket number $N_{[]}(\epsilon, \mathcal{F}, L_2(P))$ is the minimum number of $\epsilon$-brackets needed to cover $\mathcal{F}$.

**Lemma 7** ( [13], Cor. 19.35)**.** *For any class $\mathcal{F}$ of measurable functions with envelope function $F$,*

$$\mathbb{E} \left[ \sup_{f \in \mathcal{F}} |\mathbb{G}_n f| \right] \lesssim J_{[]}(\|F\|_{P,2}, \mathcal{F}, L_2(P)), \tag{38}$$

where $J_{[]}(\|F\|_{P,2}, \mathcal{F}, L_2(P))$ is called the bracketing integral:

$$J_{[]}(\|F\|_{L_2(P)}, \mathcal{F}, L_2(P)) = \int_0^{\|F\|_{L_2(P)}} \sqrt{\log\left(N_{[]}(\epsilon, \mathcal{F}, L_2(P))\right)}\, \mathrm{d}\epsilon. \tag{39}$$

**Lemma 8** (McDiarmid's Inequality, [14])**.** *Let $Z_1, \ldots, Z_n$ be real-valued independent random variables. Let $f : \mathbb{R}^n \to \mathbb{R}$ be a function that satisfies*

$$\sup_{z_1, \cdots, z_n, z_i'} \left| f(z_1, \cdots, z_{i-1}, z_i, z_{i+1}, \cdots, z_n) - f(z_1, \ldots, z_{i-1}, z_i', z_{i+1}, \cdots, z_n) \right| \le c_i,$$

*for every $i = 1, \cdots, n$. Then*

$$\mathbb{P}\left[ \left| f(Z_1, \cdots, Z_n) - \mathbb{E}\left[ f(Z_1, \cdots, Z_n) \right] \right| \ge \epsilon \right] \le 2\exp\left( -\frac{2\epsilon^2}{\sum_{i=1}^n c_i^2} \right).$$

**Lemma 9** (Vector-valued Comparison Inequality for Rademacher Process, [15], Corollary 2)**.** *Let $\mathcal{F}$ be a class of functions $\boldsymbol{f} : \mathbb{R}^D \to \mathbb{R}^c$ and let $h_i : \mathbb{R}^c \to \mathbb{R}$ be 1-Lipschitz functions. Then, for any $\boldsymbol{v}_1, \ldots, \boldsymbol{v}_N \in \mathbb{R}^D$, we have*

$$\mathbb{E}\left[ \sup_{\boldsymbol{f} \in \mathcal{F}} \sum_{i=1}^N \varepsilon_i h_i(\boldsymbol{f}(\boldsymbol{v}_i)) \right] \le \sqrt{2}\,\mathbb{E}\left[ \sup_{\boldsymbol{f} \in \mathcal{F}} \sum_{i=1}^N \boldsymbol{\varepsilon}_i^\top \boldsymbol{f}(\boldsymbol{v}_i) \right], \tag{40}$$

*where $\varepsilon_i$ are indepedent Rademacher random variables, and each $\boldsymbol{\varepsilon}_i \in \mathbb{R}^c$ is indepenent and it contains indepedent Rademacher random variables.*

**Lemma 10** (Rademacher Symmetrization, [16], Thm. 1.1)**.** *Let $\mathcal{F}$ be a class of functions $f : \mathbb{R}^D \to \mathbb{R}$ such that $0 \le f(\boldsymbol{v}) \le 1$. Let $\varepsilon_i$ be Rademacher random variables. Then for independent and identically distributed random variables $\boldsymbol{v}_1, \ldots, \boldsymbol{v}_n$, we have*

$$\mathbb{E}\left[ \sup_{f \in \mathcal{F}} \left( \frac{1}{N} \sum_{i=1}^N f(\boldsymbol{v}_i) - \mathbb{E}[f(\boldsymbol{v})] \right) \right] \le 2\mathbb{E}\left[ \sup_{f \in \mathcal{F}} \frac{1}{N} \sum_{i=1}^N \varepsilon_i f(\boldsymbol{v}_i) \right], \tag{41}$$

$$\mathbb{E}\left[ \sup_{f \in \mathcal{F}} \left( \mathbb{E}[f(\boldsymbol{v})] - \frac{1}{N} \sum_{i=1}^N f(\boldsymbol{v}_i) \right) \right] \le 2\mathbb{E}\left[ \sup_{f \in F} \frac{1}{N} \sum_{i=1}^N \varepsilon_i f(\boldsymbol{v}_i) \right]. \tag{42}$$

We also require the covering number of $\mathbb{S}(D, c)$, which could be easily derived from the standard result for the sphere. Denote by $\mathcal{N}_\epsilon$ an $\epsilon$-net of $\mathbb{S}(D, c)$ if every point $\boldsymbol{B} \in \mathbb{S}(D, c)$ can be approximated to within $\epsilon$ by some point $\boldsymbol{B}' \in \mathcal{N}_\epsilon$. The minimal cardinality of an $\epsilon$-net, denoted by $\mathcal{N}(\mathbb{S}^{D-1}, \epsilon)$, is called the covering number of $\mathbb{S}(D, c)$.

**Lemma 11.** *(Covering Number of $\mathbb{S}(D, c)$, [11, Lemma 5.2]) For every $\epsilon > 0$, the covering number of the sphere $\mathbb{S}^{D-1}$ satisfies*

$$\mathcal{N}(\mathbb{S}^{D-1}, \epsilon) \le \left( 1 + \frac{2}{\epsilon} \right)^{cD}. \tag{43}$$

We finally require one more result converning the probability that $\left\|\operatorname{sign}(\boldsymbol{o}^\top \boldsymbol{B}) - \operatorname{sign}(\boldsymbol{o}^\top \boldsymbol{B}')\right\|$ is small when $\boldsymbol{B}$ is very close to $\boldsymbol{B}'$.

**Lemma 12.** *Denote by* $\mathbb{B}(\boldsymbol{B}, \epsilon_1)$ *the set of points that around* $\boldsymbol{B}$:

$$\mathbb{B}(\boldsymbol{B}, \epsilon_1) := \left\{ \boldsymbol{B}' \in \mathbb{O}(c, D) : \|\boldsymbol{B} - \boldsymbol{B}'\|_2 \leq \epsilon_1 \right\}.$$

*Let* $\boldsymbol{o} \in \mathbb{S}^{D-1}$ *be drawn independently and uniformaly at random from the unit sphere* $\mathbb{S}^{D-1}$. *For any* $\boldsymbol{B} \in \mathbb{S}^{D-1}$ *and* $\epsilon_2 > 0$, *define*

$$\overline{\mathbb{A}} := \left\{ \boldsymbol{o} \in \mathbb{R}^D : \left\|\operatorname{sign}(\boldsymbol{o}^\top \boldsymbol{B}) - \operatorname{sign}(\boldsymbol{o}^\top \boldsymbol{B}')\right\| \leq \epsilon_2, \ \forall \ \boldsymbol{B}' \in \mathbb{B}((\boldsymbol{B}), \epsilon_1) \right\}. \tag{44}$$

*Then*

$$\mathbb{P}\left[\boldsymbol{o} \in \overline{\mathbb{A}}^c\right] \lesssim c_D D \frac{\epsilon_1^2}{\epsilon_2^2},$$

*where* $\lesssim$ *means smaller than up to a universal constant which is independent of* $D$.

*Proof.* We first bound the difference between $\operatorname{sign}(\boldsymbol{o}^\top \boldsymbol{B})$ and $\operatorname{sign}(\boldsymbol{o}^\top \boldsymbol{B}')$ by

$$
\begin{aligned}
\left\|\operatorname{sign}(\boldsymbol{o}^\top \boldsymbol{B}) - \operatorname{sign}(\boldsymbol{o}^\top \boldsymbol{B}')\right\| &= \left\| \frac{\boldsymbol{o}^\top \boldsymbol{B}}{\|\boldsymbol{o}^\top \boldsymbol{B}\|} - \frac{\boldsymbol{o}^\top \boldsymbol{B}'}{\|\boldsymbol{o}^\top \boldsymbol{B}'\|} \right\| = \left\| \frac{\|\boldsymbol{o}^\top \boldsymbol{B}'\| \boldsymbol{o}^\top \boldsymbol{B} - \|\boldsymbol{o}^\top \boldsymbol{B}\| \boldsymbol{o}^\top \boldsymbol{B}'}{\|\boldsymbol{o}^\top \boldsymbol{B}\| \|\boldsymbol{o}^\top \boldsymbol{B}'\|} \right\| \\
&= \left\| \frac{\|\boldsymbol{o}^\top \boldsymbol{B}'\| \boldsymbol{o}^\top (\boldsymbol{B} - \boldsymbol{B}') - (\|\boldsymbol{o}^\top \boldsymbol{B}\| - \|\boldsymbol{o}^\top \boldsymbol{B}'\|) \boldsymbol{o}^\top \boldsymbol{B}'}{\|\boldsymbol{o}^\top \boldsymbol{B}\| \|\boldsymbol{o}^\top \boldsymbol{B}'\|} \right\| \\
&\leq \frac{\|\boldsymbol{B} - \boldsymbol{B}'\|}{\|\boldsymbol{o}^\top \boldsymbol{B}\|} + \frac{\left|\|\boldsymbol{o}^\top \boldsymbol{B}\| - \|\boldsymbol{o}^\top \boldsymbol{B}'\|\right|}{\|\boldsymbol{o}^\top \boldsymbol{B}\|} \\
&\leq 2 \frac{\|\boldsymbol{B} - \boldsymbol{B}'\|}{\|\boldsymbol{o}^\top \boldsymbol{B}\|} \leq 2 \frac{\epsilon_1}{\|\boldsymbol{o}^\top \boldsymbol{B}\|}.
\end{aligned}
$$

Thus, as long as $\left\|\boldsymbol{o}^\top \boldsymbol{B}\right\| \geq \frac{\epsilon_1}{2\epsilon_2}$, we have $\left\|\operatorname{sign}(\boldsymbol{o}^\top \boldsymbol{B}) - \operatorname{sign}(\boldsymbol{o}^\top \boldsymbol{B}')\right\| \leq \epsilon_2$. Without loss of generality, suppose $\boldsymbol{B} = \begin{bmatrix} \boldsymbol{e}_1 & \cdots \boldsymbol{e}_c \end{bmatrix}$. Then, the probability that $\boldsymbol{o} \in \overline{\mathbb{A}}^c$ is bounded by the probability that $\left\|\boldsymbol{o}^\top \boldsymbol{B}\right\| \leq \frac{\epsilon_1}{2\epsilon_2}$:

$$\mathbb{P}\left[\boldsymbol{o} \in \overline{\mathbb{A}}^c\right] \leq \mathbb{P}\left[\left\|\boldsymbol{o}^\top \boldsymbol{B}\right\| \leq \frac{\epsilon_1}{2\epsilon_2}\right] \leq \mathbb{P}\left[o_1 \leq \frac{\epsilon_1}{2\epsilon_2}\right] \lesssim c_D D \frac{\epsilon_1^2}{\epsilon_2^2}.$$

where $o_1$ is the first element in $\boldsymbol{o}$, and the last inequality follows from [10, Lemma 12]. $\qquad\square$

## 10.2   Proof of Lemma 5

Before givin out the main proofs, we first preset the following useful result concerning the expectation of $\eta_{\boldsymbol{\mathcal{O}}}$.

**Lemma 13.** *Suppose* $\boldsymbol{o}_1, \cdots, \boldsymbol{o}_M$, *are drawn independently and uniformly at random from the unit sphere* $\mathbb{S}^{D-1}$. *Then*

$$\mathbb{E}\left[ \sup_{\boldsymbol{B}, \boldsymbol{G} \in \mathbb{O}(c, D), \boldsymbol{G} \perp \boldsymbol{B}} \left| \sum_{j=1}^{M} \left\langle \operatorname{sign}(\boldsymbol{B}^\top \boldsymbol{o}_j), \boldsymbol{G}^\top \boldsymbol{o}_j \right\rangle \right| \right] \lesssim \sqrt{cD} \log\left(\sqrt{c_D D}\right) \sqrt{M}, \tag{45}$$

where $\lesssim$ means smaller than up to a universal constant which is independent of $D$ and $M$.

*Proof.* The main idea for proving Lemma 13 is to view

$$\frac{1}{\sqrt{M}} \sup_{\boldsymbol{B},\boldsymbol{G}\in\mathbb{O}(c,D),\boldsymbol{G}\perp\boldsymbol{B}} \left| \sum_{j=1}^{M} \left\langle \text{sign}(\boldsymbol{B}^\top \boldsymbol{o}_j), \boldsymbol{G}^\top \boldsymbol{o}_j \right\rangle \right|$$

as an empirical process and then utilize Lemma 7. Towards that end, define the set

$$\mathbb{F} := \{(\boldsymbol{B}, \boldsymbol{B}) : \boldsymbol{B}, \boldsymbol{G} \in \mathbb{O}(c, D), \boldsymbol{G} \perp \boldsymbol{B}\}.$$

We further define the parameterized function as

$$f_{\boldsymbol{B},\boldsymbol{G}}(\boldsymbol{o}) := \left\langle \text{sign}(\boldsymbol{B}^\top \boldsymbol{o}), \boldsymbol{G}^\top \boldsymbol{o} \right\rangle.$$

The class of functions we are interested in is $\mathcal{F} := \{f_{\boldsymbol{B},\boldsymbol{G}} : (\boldsymbol{B},\boldsymbol{G}) \in \mathbb{F}\}$.

Note that for any $f_{\boldsymbol{B},\boldsymbol{G}} \in \mathcal{F}$ (i.e., $(\boldsymbol{B},\boldsymbol{G}) \in \mathbb{F}$), we have

$$\mathbb{E}\left[f_{\boldsymbol{B},\boldsymbol{G}}(\boldsymbol{o})\right] = \mathbb{E}\left[\left\langle \text{sign}(\boldsymbol{B}^\top \boldsymbol{o}), \boldsymbol{G}^\top \boldsymbol{o} \right\rangle\right] = 0,$$

which together with (37) indicates that

$$\sum_{j=1}^{M} \left\langle \text{sign}(\boldsymbol{B}^\top \boldsymbol{o}), \boldsymbol{G}^\top \boldsymbol{o} \right\rangle = \sqrt{M} \mathbb{G}_M f_{\boldsymbol{B},\boldsymbol{G}},$$

where $\mathbb{G}_M f_{\boldsymbol{B},\boldsymbol{G}}$ is the empirical process of $f_{\boldsymbol{B},\boldsymbol{G}}$.

To utilize Lemma 7, the rest of the proof is to show the corresponding bracketing integral is finite for our problem. Since $|f_{\boldsymbol{B},\boldsymbol{G}}(\boldsymbol{o})| \leq \|\boldsymbol{o}\|_2$ for any $(\boldsymbol{B},\boldsymbol{G}) \in \mathbb{F}$, we know $F(\boldsymbol{o}) = \|\boldsymbol{o}\|_2$ is the envelope function of $\mathcal{F}$ and $\|F\|_{P,2} = 1$. Thus, we only need to consider the the bracket integral $J_{[]}(1, \mathcal{F}, L_2(P))$, where $P$ is now a probability measure on $\mathbb{B}(\boldsymbol{B},\boldsymbol{G})$. To that end, we first compute the bracket number $N_{[]}(\epsilon, \mathcal{F}, L_2(P))$.

Since our function $f_{\boldsymbol{B},\boldsymbol{G}}$ is parameterized by $(\boldsymbol{B},\boldsymbol{G})$, covering the class of functions $\mathcal{F}$ is related to covering the set $\mathbb{F}$. For any fixed $(\boldsymbol{B},\boldsymbol{G}) \in \mathbb{F}$, define the set of points that around $(\boldsymbol{B},\boldsymbol{G})$:

$$\mathbb{B}((\boldsymbol{B},\boldsymbol{G}),\epsilon_1) := \left\{ (\boldsymbol{B}',\boldsymbol{G}') \in \mathbb{F} : \sqrt{\|\boldsymbol{B}-\boldsymbol{B}'\|_F^2 + \|\boldsymbol{G}-\boldsymbol{G}'\|_F^2} \leq \epsilon_1 \right\}.$$

Then, denote by

$$\mathbb{A} := \left\{ \boldsymbol{o} \in \mathbb{R}^D : \left\|\text{sign}(\boldsymbol{o}^\top \boldsymbol{B}) - \text{sign}(\boldsymbol{o}^\top \boldsymbol{B}')\right\| \leq \epsilon_2, \ \forall \ (\boldsymbol{B}',\boldsymbol{G}') \in \mathbb{B}((\boldsymbol{B},\boldsymbol{G}),\epsilon_1) \right\}.$$

When $\boldsymbol{B}$ is close to $\boldsymbol{B}'$, then $\mathbb{A}$ should cover most of $\boldsymbol{o}$. If $\boldsymbol{o} \in \mathbb{A}$, then for any $(\boldsymbol{B}',\boldsymbol{G}') \in \mathbb{B}((\boldsymbol{B},\boldsymbol{G}),\epsilon_1)$ we have

$$\begin{aligned}
|f_{\boldsymbol{B},\boldsymbol{G}}(\boldsymbol{o}) - f_{\boldsymbol{B}',\boldsymbol{G}'}(\boldsymbol{o})| &= \left| \left\langle \text{sign}(\boldsymbol{B}^\top \boldsymbol{o}), \boldsymbol{G}^\top \boldsymbol{o} \right\rangle - \left\langle \text{sign}(\boldsymbol{B}'^\top \boldsymbol{o}), \boldsymbol{G}'^\top \boldsymbol{o} \right\rangle \right| \\
&= \left| \left\langle \text{sign}(\boldsymbol{B}^\top \boldsymbol{o}), (\boldsymbol{G}-\boldsymbol{G}')^\top \boldsymbol{o} \right\rangle - \left\langle \left(\text{sign}(\boldsymbol{B}'^\top \boldsymbol{o}) - \text{sign}(\boldsymbol{B}^\top \boldsymbol{o})\right), \boldsymbol{G}'^\top \boldsymbol{o} \right\rangle \right| \\
&\leq \left\|\boldsymbol{G}-\boldsymbol{G}'\right\| + \left\|\text{sign}(\boldsymbol{B}'^\top \boldsymbol{o}) - \text{sign}(\boldsymbol{B}^\top \boldsymbol{o})\right\| \\
&\leq \epsilon_1 + \epsilon_2.
\end{aligned}$$

On the other hand, if $\boldsymbol{o} \in \mathbb{A}^c$, then for any $(\boldsymbol{B}', \boldsymbol{G}') \in \mathbb{B}((\boldsymbol{B}, \boldsymbol{G}), \epsilon_1)$ we have

$$|f_{\boldsymbol{B},\boldsymbol{G}}(\boldsymbol{o}) - f_{\boldsymbol{B}',\boldsymbol{G}'}(\boldsymbol{o})| = \left|\left\langle \text{sign}(\boldsymbol{B}^\top \boldsymbol{o}), \boldsymbol{G}^\top \boldsymbol{o} \right\rangle\right| + \left|\left\langle \text{sign}(\boldsymbol{B}'^\top \boldsymbol{o}), \boldsymbol{G}'^\top \boldsymbol{o} \right\rangle\right| \leq 2.$$

To summary, we have

$$|f_{\boldsymbol{B},\boldsymbol{G}}(\boldsymbol{o}) - f_{\boldsymbol{B}',\boldsymbol{G}'}(\boldsymbol{o})| \leq \epsilon_1 \delta_{\mathbb{A}}(\boldsymbol{o}) + 2\delta_{\mathbb{A}^c}(\boldsymbol{o}), \ \forall \ (\boldsymbol{B}', \boldsymbol{G}') \in \mathbb{B}((\boldsymbol{B}, \boldsymbol{G}), \epsilon_1). \tag{46}$$

We now define a bracket $[l, u]$ by

$$l(\boldsymbol{o}) = f_{\boldsymbol{B},\boldsymbol{G}}(\boldsymbol{o}) - \epsilon_1 \delta_{\mathbb{A}}(\boldsymbol{o}) - 2\delta_{\mathbb{A}^c}(\boldsymbol{o}),$$
$$u(\boldsymbol{o}) = f_{\boldsymbol{B},\boldsymbol{G}}(\boldsymbol{o}) + \epsilon_1 \delta_{\mathbb{A}}(\boldsymbol{o}) + 2\delta_{\mathbb{A}^c}(\boldsymbol{o}),$$

where the indicator function $\delta_{\mathbb{A}}(\boldsymbol{o})$ is defined as $\delta_{\mathbb{A}}(\boldsymbol{o}) = \begin{cases} 1, & \boldsymbol{o} \in \mathbb{A} \\ 0, & \boldsymbol{o} \in \mathbb{A}^c \end{cases}$. Due to (46), we have $f_{\boldsymbol{B}',\boldsymbol{G}'} \in [l, u]$ for all $(\boldsymbol{B}', \boldsymbol{G}') \in \mathbb{B}((\boldsymbol{B}, \boldsymbol{G}), \epsilon_1)$. Also,

$$\|u - l\|_{L_2(P)} = \|2\epsilon_1 \delta_{\mathbb{A}}(\boldsymbol{o}) + 4\delta_{\mathbb{A}^c}(\boldsymbol{o})\|_{L_2(P)} = \sqrt{4(\epsilon_1 + \epsilon_2)^2 \mathbb{P}[\boldsymbol{o} \in \mathbb{A}] + 16\mathbb{P}[\boldsymbol{o} \in \mathbb{A}^c]}$$
$$< 2(\epsilon_1 + \epsilon_2) + 4\sqrt{\mathbb{P}[\boldsymbol{o} \in \mathbb{A}^c]} \leq 2(\epsilon_1 + \epsilon_2) + 4\sqrt{c_1 c_D D} \frac{\epsilon_1}{\epsilon_2}, \tag{47}$$

and the last inequality follows because $\mathbb{P}[\boldsymbol{o} \in \mathbb{A}^c] \leq c_1 c_D D \epsilon_1^2$ according to Lemma 12 with $c_1$ a universal constant

Finally, the number of brackets to cover $\mathcal{F}$ is equal to the number of such balls $\mathbb{B}((\boldsymbol{B}, \boldsymbol{G}), \epsilon_1)$ that cover $\mathbb{F}$. Utilizing Lemma 11, the covering number for $\mathbb{F}$ is

$$\mathcal{N}(\mathbb{F}, \epsilon_1) \leq \left(1 + \frac{2\sqrt{2}}{\epsilon_1}\right)^{2cD}. \tag{48}$$

Recall the definition that the bracket number $N_{[]}(\epsilon, \mathcal{F}, L_2(P))$ is the minimum number of $\epsilon$-brackets needed to cover $\mathcal{F}$, where an $\epsilon$-bracket in $L_2(P)$ is a bracket $[l, u]$ with $\|u - l\|_{L_2(P)} \leq \epsilon$. Thus, by letting $\epsilon_2 = \sqrt{\epsilon_1}$ and $2(\epsilon_1 + \sqrt{\epsilon_2}) + 4\sqrt{c_1 c_D D}\sqrt{\epsilon_1} = \epsilon$ and plugging this into (48), we obtain the bracket number

$$N_{[]}(\epsilon, \mathcal{F}, L_2(P)) \leq \left(1 + c_2 \frac{c_D D}{\epsilon^2}\right)^{2cD},$$

where $c_2$ is a universal constant. Now plug this into Lemma 7 gives

$$\frac{1}{\sqrt{M}} \mathbb{E}\left[\sup_{\boldsymbol{B},\boldsymbol{G} \in \mathbb{O}(c,D), \boldsymbol{G} \perp \boldsymbol{B}} \left|\sum_{j=1}^{M} \left\langle \text{sign}(\boldsymbol{B}^\top \boldsymbol{o}_j), \boldsymbol{G}^\top \boldsymbol{o}_j \right\rangle\right|\right] \lesssim \int_0^1 \sqrt{\left(1 + c_2 \frac{c_D D}{\epsilon^2}\right)^{2cD}} \, \mathrm{d}\epsilon$$
$$\lesssim \sqrt{cD} \log(c_D D).$$

$\square$

We are now ready to prove Lemma 5. Let $\boldsymbol{o}'_k$ be any points of $\mathbb{S}^{D-1}$. Since the product of compact spaces is compact, there exist $\boldsymbol{B}^*, \boldsymbol{G}^* \in \mathbb{O}(c, D)$ for which the value $\sup_{\boldsymbol{B},\boldsymbol{G} \in \mathbb{S}^{D-1}, \boldsymbol{B} \perp \boldsymbol{g}} |\sum_{j=1}^{M} \text{sign}(\boldsymbol{B}^\top \boldsymbol{o}_j) \boldsymbol{g}^\top \boldsymbol{o}_j|$ is achieved. Then, we have

$$\left| \sup_{\boldsymbol{B},\boldsymbol{G} \in \mathbb{O}(c,D), \boldsymbol{G} \perp \boldsymbol{B}} \left| \sum_{j=1}^{M} \left\langle \text{sign}(\boldsymbol{B}^\top \boldsymbol{o}_j), \boldsymbol{G}^\top \boldsymbol{o}_j \right\rangle \right| \right. \tag{49}$$

$$\left. - \sup_{\boldsymbol{B},\boldsymbol{G} \in \mathbb{O}(c,D), \boldsymbol{G} \perp \boldsymbol{B}} \left| \sum_{j \neq k} \left\langle \text{sign}(\boldsymbol{B}^\top \boldsymbol{o}_j), \boldsymbol{G}^\top \boldsymbol{o}_j \right\rangle + \left\langle \text{sign}(\boldsymbol{B}^\top \boldsymbol{o}'_k), \boldsymbol{G}^\top \boldsymbol{o}'_k \right\rangle \right| \right| \tag{50}$$

$$\leq \left\| \left| \sum_{j=1}^{M} \left\langle \text{sign}(\boldsymbol{B}^{*\top} \boldsymbol{o}_j), \boldsymbol{G}^{*\top} \boldsymbol{o}_j \right\rangle \right| - \left| \sum_{j \neq k} \left\langle \text{sign}(\boldsymbol{B}^{*\top} \boldsymbol{o}_j), \boldsymbol{G}^{*\top} \boldsymbol{o}_j \right\rangle + \left\langle \text{sign}(\boldsymbol{B}^{*\top} \boldsymbol{o}'_k), \boldsymbol{G}^{*\top} \boldsymbol{o}'_k \right\rangle \right| \right\| \tag{51}$$

$$\leq \left| \left\langle \text{sign}(\boldsymbol{B}^{*\top} \boldsymbol{o}_k), \boldsymbol{G}^{*\top} \boldsymbol{o}_k \right\rangle - \left\langle \text{sign}(\boldsymbol{B}^{*\top} \boldsymbol{o}_k), \boldsymbol{G}^{*\top} \boldsymbol{o}'_k \right\rangle \right| \leq 2, \tag{52}$$

where the second inequality follows from the reverse triangle inequality. Applying Lemma 8 with $c_k = 2$ and using Lemma 13, we obtain

$$\mathbb{P} \left[ \sup_{\boldsymbol{B},\boldsymbol{G} \in \mathbb{O}(c,D), \boldsymbol{G} \perp \boldsymbol{B}} \left| \sum_{j=1}^{M} \left\langle \text{sign}(\boldsymbol{B}^\top \boldsymbol{o}_j), \boldsymbol{G}^\top \boldsymbol{o}_j \right\rangle \right| \gtrsim \sqrt{cD} \log\left(c_D D\right) \sqrt{M} + \epsilon \right] \leq 2 \exp\left( -\frac{2\epsilon^2}{4M} \right). \tag{53}$$

Finally, set $\epsilon = t\sqrt{M}$ to get

$$\mathbb{P} \left[ \sup_{\boldsymbol{B},\boldsymbol{G} \in \mathbb{O}(c,D), \boldsymbol{G} \perp \boldsymbol{B}} \left| \sum_{j=1}^{M} \left\langle \text{sign}(\boldsymbol{B}^\top \boldsymbol{o}_j), \boldsymbol{G}^\top \boldsymbol{o}_j \right\rangle \right| \gtrsim \left( \sqrt{cD} \log\left(c_D D\right) + t \right) \sqrt{M} \right] \leq 2 \exp(-t^2/2). \tag{54}$$

## Footnotes

[1] For the case $c > d$, we have at least $\phi_1 = \cdots = \phi_{c-d} = 0$. Similar to the case $c \leq d$, we can also rewrite $\boldsymbol{S}^\top \boldsymbol{B} = \boldsymbol{V}\sin(\boldsymbol{\Phi})\boldsymbol{R}^\top$, where $\boldsymbol{V} = \begin{bmatrix} \mathbf{0} & \overline{\boldsymbol{V}} \end{bmatrix}$ with $\overline{\boldsymbol{V}} \in \mathbb{R}^{d \times d}$ an orthonormal matrix. Thus, we also have (24) and the following proofs are the same.