[Reviews · NeurIPS 2019]

Reviewer 1



I am not an expert on the Riemannian optimization. At first glance, this work seems to be similar to [11]. I wonder what makes it nontrivial to extend the regularity condition (sharpness and weak convexity) and the proof technique in [11] to Riemmanian optimization.

Reviewer 2



A number of problems in sparse learning, signal processing, etc., can be phrased as optimizing a nonsmooth function over a riemannian manifold. Many works avoid nonsmooth analysis / optimization, by applying smooth methods to a smoothing of the objective function, often at the cost of suboptimalities in convergence rate, sample complexity, etc.. This work takes a different path, directly developing methods for nonsmooth riemannian optimization. The focus on the grassmannian limits the scope to some extent. It is unclear what in the setup requires the grassmannian. The regularity condition seems like it can be extended to arbitrary submanifolds of Euclidean space. Intuitively, one would expect the same projected subgradient method to succeed; one could potentially also replace the projection with any second order retraction. Is there anywhere where the paper uses the special structure of the grassmannian, aside from working out applications in section 4? There is a natural intrinsic notion of a riemannian subgradient, in which one takes $D$ in the tangent space, and the condition becomes $\lim \inf_{A \to B} \frac{f(A) - f(B) - < D, log_B(A) - B >}{ d(B,A) } >= 0$ (see, e.g., reference [17]). In the smooth case, this coincides with the the definition (4): the the riemannian gradient is the projection of the euclidean gradient onto the tangent space. In the nonsmooth case, this intrinsic notion can differ from (4). It would be helpful to clarify the assumptions that are needed to ensure that (4) indeed the Riemannian sub differential. The paper discusses the regularity condition as a generalization of sharpness and weak convexity to the Grassmannian. This paragraph is imprecisely phrased: both sharpness and weak convexity already have very natural Riemannian analogues. The RRC here is not a generalization of sharpness and weak convexity, but rather of the consequence (9). Again, it is not clear that this condition has been phrased in an intrinsic manner; calling it a Riemannian regularity condition may introduce confusion. These definitional issues do not affect the correctness of the paper’s main claims, since once it is assumed that the regularity condition holds with the objects employed in (4), the paper’s claims follow. The experiments support the main claims of the paper. The proposed method indeed converges linearly with appropriately chosen geometrically decreasing step size. The paper does a good job of working out theoretical consequences of its claims for specific models. The results on dictionary learning follow from a combination of the general theoretical work done here and analysis in [2]; the results on DPCP require additional work to verify the regularity condition, which is a technical contribution of the submission. EDIT: The authors response has done a good job of addressing the above issues with definitions and terminology. I have updated my score to an 8.

Reviewer 3



I have read the authors' response and believe it adequately addresses all reviewers comments. -------------- Original Comments: The authors present a regularity condition with proof that it guarantees linear convergence of projected Grassmannian subgradient descent. The regularity condition is intuitive and useful in that it only requires characterization of the Riemannian subgradient in the neighborhood of the desired point, and potentially applies to a wide class of functions. In addition, they show that their analysis holds for two problems of modern interest (Dual Principal Component Pursuit and Orthogonal Dictionary Learning), leading to new convergence results for these problems. For my part, I have verified the correctness of Proposition 1, and Theorem 1. Specific Notes: Line 84: I found the definition a bit off-putting at first for the obvious reason that we usually think of the projection of B onto A as an operation on B. I recognize that in this setting if A and B are elements of O(c, D) then the definition is equivalent. I just think it may be nice to include a sentence stating this for clarity. Line 116: neighborhood Line 118: Is it more appropriate to say this gives abound on the sum of the cosines of the principal angles? Section 3.2: Although it is clear what you mean, it may be better to explicitly state that by projection onto the Grassmannian you mean orthonormalization. Alternatively (and perhaps preferably), you could formally define projection onto the Grassmannian. General Notes: It seems to me that the convergence results should hold even if the steps are taken along a geodesic. I have not explored this idea very much, but it may be straight forward to extend similar results to non-projected Grassmannian gradient descent.

[Author Response · NeurIPS 2019]

We thank all three reviewers for their constructive comments. We address them below one by one.

**To Reviewer 1 (R1).**

Q1: *what makes it nontrivial to extend the regularity condition and proof technique in [11] to Riemmanian optimization.*

A1: The key contribution of our work is not to extend sharpness and weak convexity to the Riemannian regularity
condition (RRC), but rather to use the RRC to analyze the convergence of the projected Riemannian subgradient method
(PRSM) and to demonstrate the applicability of the RRC to problems in robust subspace and dictionary learning. The
result in [11] only works for convex sets that have a nice property such as their orthogonal projector being non-expansive.
The Grassmannian manifold is nonconvex, making the analysis more complex. We have exploited specific structure in
the Grassmannian and used a Riemannian subgradient instead of subgradient (which is used in [11]) to get our results.

Q2: *For DPCP, the contribution can be bigger if a global convergence guarantee is provided using some initialization*
*(e.g. the spectral initialization analyzed in the appendix)*

A2: We do have a deterministic and a statistical analysis for the spectral initialization in the supplementary material;
see Lemma 3 and Corollary 3, which are not mentioned in the manuscript. In particular, Corollary 3 implies that in a
random model, the spectral method provides a valid initialization with high probability when the number of outliers is
smaller than the square of the number of inliers. We will incorporate these into a revised version of the manuscript.

**To Reviewer 3 (R3).**

Q1: *The focus on the grassmannian limits the scope to some extent. The regularity condition seems like it can be extended*
*to arbitrary submanifolds of Euclidean space. Is there anywhere using the special structure of the grassmannian?*

A1: We agree the definition of the RRC can be extended to arbitrary submanifolds of a Euclidean space with an
appropriate definition of the Riemanniann metric and distance. However, using the RRC to analyze convergence of
optimization methods requires exploiting the specific properties of the submanifold. In our paper  the Grassmannian
structure is utilized together with the RRC to analyze the convergence of the projected Riemannian subgradient
method. Specifically, an important property of the Grassmannian manifold is that if a point $B \in \mathbb{R}^{D \times c}$ is outside the
Grassmannian (i.e., $\sigma_c(B) \geq 1$), then projecting it onto the Grassmannian will not increase its distance to any other
point in the Grassmannian with the distance defined in (2). If other manifolds also have such property, then the current
convergence analysis can also be applied. We will incorporate this discussion in the final version.

Q2: *It would be helpful to clarify the assumptions that are needed to ensure that (4) is the Riemannian subdifferential.*

A2: We really appreciate this comment. The reviewer is correct that for a general nonsmooth function, the current (4)
may not be the Riemannian subdifferential. To be more rigorous, in the revision we will directly define the Riemannian
subdifferential using the Clarke subdifferential (which is more general than the Fréchet subdifferential) for locally
Lipschitz functions on Riemannian manifolds [2]. As pointed out by the reviewer, the analysis still holds as long as the
Riemannian regularity condition holds. According to [A], if a function is **regular**, then the Riemannian subdifferential
based on the Clark subdifferential is equal to the projection (onto the tangent space) of the Clark subdifferential. Since
both the robust subspace learning and dictionary learning problems are regular, their Riemannian subdifferentials
computed in Section 4 are correct. We will incorporate this discussion in the revision.

[A] Yang, Zhang, Song. "Optimality conditions for the nonlinear programming problems on Riemannian manifolds."
Pacific Journal of Optimization, 2014.

Q3: *The RRC here is not a generalization of sharpness and weak convexity, but rather of the consequence (9).*

A3: The referee is correct, and we will rephrase this sentence in the final version, if accepted.

Q4: *It is not clear that this condition has been phrased in an intrinsic manner; calling it RRC may introduce confusion.*

A4: We called our condition a Riemannian Regularity Condition because it involves the Riemannian subgradient.
However, we agree that this definition is extrinsic. We will rename our definition as Extrinsic RRC to avoid confusion.

**To Reviewer 4 (R4).**

Q1: *Line 84: we usually think of the projection of $B$ onto $A$ as an operation on $B$.*

A1: Since we project $B$ onto $[A]$, here $AQ^\star$ represents a point in $[A]$. Of course, the reviewer is correct that $Q^\star$
contains a nonlinear transformation of $A^\top B$. We will incorporate this discussion into the final version, if accepted.

Q2: *Line 118: Is it more appropriate to say this gives a bound on the sum of the cosines of the principal angles?*

A2: This is a great suggestion and we will incorporate it into a revision.

Q3: *Section 3.2: it maybe better to explicitly state that by projection onto Grassmannian you mean orthonormalization.*

A3: This is a great suggestion. We will make this statement in a revision of the paper.

Q4: *It seems to me that the convergence results should hold even if the steps are taken along a geodesic.*

A4: In this paper we take an extrinsic approach because extrinsic methods are typically easier to implement, e.g. when
the projection map is easier to compute than the geodesic distance. Extending the current analysis to an intrinsic
optimization method is definitely worth exploring and will be the subject of future research.

[Meta-Review · NeurIPS 2019]

The reviews as well as the author response built a convincing case for this paper. The remaining concerns are listed in the reviews, and the authors should apply those when preparing camera-ready.